# Aging-related peroxisomal dysregulation disrupts intestinal stem cell differentiation through alterations of very long-chain fatty acid oxidation

Xiaoxin Guo[1,2,☉], Gang Du[3,4☉], Juanyu Zhou[1☉], Fang Fu[1], Yu Yuan[1], Xingzhu Liu[1], Haiou Chen[1], Qianyi Wan[1,5], Bo Gong[2], Haiyang Chen [1]*

1 West China Centre of Excellence for Pancreatitis and Laboratory of Metabolism and Aging Research, National Clinical Research Center for Geriatrics, West China Hospital, Sichuan University, Chengdu, Sichuan, China, 2 Genetic Diseases Key Laboratory of Sichuan Province, Department of Medical Genetics, Department of Laboratory Medicine, Sichuan Academy of Medical Sciences & Sichuan Provincial People's Hospital, School of Medicine, University of Electronic Science and Technology of China, Chengdu, Sichuan, China, 3 Department of Biological Chemistry and Molecular Pharmacology, Harvard Medical School, Boston, Massachusetts, United States of America, 4 Program in Cellular and Molecular Medicine, Boston Children's Hospital, Boston, Massachusetts, United States of America, 5 Division of Gastrointestinal Surgery, Department of General Surgery, West China Hospital, Sichuan University, Chengdu, Sichuan, China

☉ These authors contributed equally to this work.
* chenhy82@scu.edu.cn

## Abstract

Aging disrupts intestinal stem cell (ISC) lineage fidelity, impairing epithelial barrier function and then promoting systemic health decline. In this study, we identify peroxisomal dysfunction as a critical driver of age-associated ISC mis-differentiation. Using *Drosophila* and mouse colonic organoids, we demonstrate that reduced PEX5 expression in aged ISCs impairs peroxisomal matrix protein import, leading to very long-chain fatty acids (VLCFAs) accumulation. In addition, we found that RAB7-dependent late endosome maturation and SOX21A were downstream of the peroxisome in controlling aged ISC differentiation. Aspirin, a classic anti-inflammatory drug, restores ISC lineage fidelity by enhancing PEX5-mediated peroxisomal β-oxidation of VLCFAs. Taken together, these findings highlight peroxisomal dysfunction and VLCFA metabolism as pivotal regulators of ISC aging and suggest new therapeutic strategies for combating age-related intestinal decline.

## Introduction

Tissue homeostatic maintenance and regenerative responsiveness to injury depend on tissue-specific stem cells [1]. Adult ISCs, which can proliferate, self-renew (produce new stem cells), or differentiate into absorptive or secretory cells, are regarded as reserve cells that support both intestinal turnover and the regenerative response

**Data availability statement:** The mass spectrometry proteomics data have been deposited to the ProteomeXchange Consortium (https://www.proteomexchange.org/) via the iProX partner repository with the dataset identifier PXD056574. The RNA-seq data that support the findings of this study have been deposited in the Sequence Read Archive (BioProject: PRJNA1169934, https://www.ncbi.nlm.nih.gov/bioproject/PRJNA1169934). The raw sequencing data of free fatty acids by mass spectrometry related to this manuscript is available in S1 File.

**Funding:** This work was supported by the National Natural Science Foundation of China (32470879) (H.C.), the National Key R&D Program of China (2020YFA0803602) (H.C.), the National Natural Science Foundation of China (82501268) (X.G.), Sichuan Science and Technology Program, Sichuan Provincial Natural Science Foundation (2025ZNSFSC0720) (H.C.), and the Noncommunicable Chronic Diseases-National Science and Technology Major Project (2023ZD0506800) (H.C.). The Project of China Postdoctoral Science Foundation (2025M772611) (X.G.). The fund for Sichuan Provincial People's Hospital (24QNPY030) (X.G.). The funders had no role in study design, data collection and analysis, decision to publish, or preparation of the manuscript.

**Competing interests:** The authors have declared that no competing interests exist.

**Abbreviations:** ACOX1, acyl-CoA oxidase 1; ALD, adrenoleukodystrophy; BA, Behenic acid; BSA, bovine serum albumin; CPF, Crude Peroxisomal Fraction; CRC, colorectal cancer; EBs, enteroblasts; EC, enterocyte; FFAs, free fatty acids; GC–MS, Gas Chromatography-Mass Spectrometry; H&E, Hematoxylin and eosin; ISC, intestinal stem cell; LFQ, label-free quantification; MS, mass spectrometry; PCA, Principal Component Analysis; Pex5-HA, Peroxin 5-HA; PTS1, peroxisomal targeting signal 1; PVDF, polyvinylidene difluoride; RNAi, RNA interference; RNA-seq, RNA sequencing; TEM, transmission electron microscopy; VLCFAs, very long-chain fatty acids; ZS, Zellweger syndrome.

following aging or injuries [2]. Aged ISCs have less proliferative and regenerative capacity, and are accompanied by an accumulation of mis-differentiated cells [3–6]. This decline in stem cell function may lead to age-related morbidities such as cancer, inflammatory disorders, diabetes, and degenerative disease. It is reported that upon aging, the alteration of cell composition in intestine is mainly due to the mis-regulation of ISC lineage fidelity [7], which manifests Paneth cells and Goblet cells increased in the mammalian intestine during aging [3,5]. However, the precise mechanisms by which aging affects ISC differentiation and leads to intestinal functional declines are still not well-understood.

Peroxisomes are distinct subcellular organelles present in all groups of eukaryotes which are involved in a variety of physiological processes. Regarded as redundant organelles, peroxisomes have historically received less attention than other organelles. However, peroxisomes participate in various metabolic pathways involving reactive oxygen species metabolism, β-oxidation of very long-chain fatty acids (VLCFAs), bile acid metabolism, and synthesis of ether lipids such as plasmalogens [8–10]. Meanwhile, dysfunction in peroxisome biogenesis can lead to various diseases, such as Zellweger syndrome (ZS), neonatal adrenoleukodystrophy, infantile Refsum disease, and pseudo-ZS, which are associated with severe developmental defects [10]. As an organelle of emerging interest, the peroxisome has been reported to participate in diverse cellular responses including oxidative stress [11], oxygen tension [12], sound exposure [13], and microbial infection [14]. In our previous findings, we found an increased number of peroxisomes in ISCs of the injured guts including human, mouse, and *Drosophila*, which then accelerated ISC-mediated intestinal epithelial repair [15]. Furthermore, we demonstrated that VLCFAs, which act as ISC niche signals, promote an increase in peroxisome abundance within ISCs. And we uncovered a feedback mechanism involving the transcription factors peroxisome proliferator-activated receptors and SOX21, which precisely regulate peroxisome dynamics [16]. However, the connection between peroxisomes and the age-related deterioration of ISC function remains largely unexplored.

Fatty acids are among the simplest lipids and can be categorized based on their carbon chain length; they include short-chain fatty acids, medium-chain fatty acids, long-chain fatty acids, and VLCFAs. VLCFAs, which contain ≥20 carbons, are specifically β-oxidized in peroxisomes [17,18]. Despite being minor components in almost all organisms, VLCFAs play many important roles, which include their role in stabilizing highly curved membranes [19], facilitating the execution of cell division [20], and promoting autophagosome formation. Contrarily, VLCFAs are also involved in inducing tumor growth [21], disrupting the cell membrane during necroptosis [22], inducing the occurrence of colorectal cancer (CRC) [23], causing X-linked adrenoleukodystrophy (ALD), and so on [10]. It has also been reported that the degradation of shorter fatty acids (those containing <20 carbons) by mitochondrial fatty acid oxidation was reduced, which weakened stem cell function during aging [4]. However, the role of VLCFAs in regulating ISC aging remains largely unknown, making it a topic deserving of in-depth study.

Many studies have shown that certain natural compounds can prevent the age-related decline in stem cell function and help maintain tissue homeostasis [24–26]. Similarly to the mammalian intestine, the *Drosophila* intestine serves as a powerful model system for studying aged ISCs and facilitates the screening of anti-aging small molecule drugs. In this study, we demonstrated that Aspirin rescues the mis-differentiation of aged ISCs by enhancing peroxisome function. PEX5 levels were decreased in aged ISCs, leading to peroxisomal dysfunction and the accumulation of VLCFAs, which ultimately resulted in the mis-differentiation of aged ISCs. Aspirin mitigated the age-associated decline of ISCs by enhancing PEX5-mediated peroxisomal function. Thus, our findings suggest that enhancing peroxisome function could be a novel target for anti-aging strategies in ISCs.

## Results

### Aspirin restrains age-associated ISC mis-differentiation and prevents intestine functional decline

Intestinal aging is characterized by alterations in villus length, crypt size, and cell composition within aged crypts, ultimately leading to the development of epithelial dysplasia and barrier dysfunction. These changes are primarily attributed to the mis-regulation of ISC lineage fidelity [7]. It has been reported that aging results in an increase in the number of secretory lineage cells in mammals [3]. In *Drosophila*, ISCs give rise to the absorptive enterocyte (EC) and secretory enteroendocrine lineages. Similarly, a continuous accumulation of progenitor cell enteroblasts (EBs) (*esg*+ cells) [27] and pre-ECs (Pdm1+ and *esg*+ cells) has also been found in aged *Drosophila* (Fig 1A and 1C–1E). To screen compounds that can inhibit age-associated differentiation defects within ISCs, we conducted a comprehensive screening of 31 small-molecule anti-aging compounds. And we found that the ratios of progenitor cells and pre-ECs were significantly decreased after Aspirin treatment in aged *Drosophila* midguts, which demonstrates that Aspirin is an effective anti-aging compound (Fig 1B–1E and S1A Fig).

Previous studies have shown that the differentiation defects of ISCs during aging cause a significant decrease in the digestive functions of *Drosophila*, including the loss of gastrointestinal acid–base homeostasis, a decline in excretion, and a loss of barrier function [24–26,28]. Since Aspirin administration can promote age-related ISC differentiation, the effects of Aspirin administration on improving the digestive functions of aged flies were further tested. Aspirin treatment starting at an intermediate age significantly alleviated the reduction of excretion (Fig 1F and S1B Fig), prevented the further deterioration of gastrointestinal acid–base homeostasis (Fig 1G and S1C Fig), and increased the intestinal barrier function (Fig 1H and S1D Fig) in aged flies. In addition, our results demonstrated that aspirin prolongs the life span of *Drosophila* when feeding begins 20 days after eclosion (Fig 1I and S1E Fig). These data suggested that Aspirin could prevent the ISC aging-induced functional decline of gastrointestinal tracks.

To investigate the possibility that Aspirin plays a role in alleviating ISC differentiation defects in mammals, a three-dimensional mouse intestinal crypt organoid culture system was prepared in our study. Colon crypts were isolated from the intestine of both young and aging mice, and treated with or without Aspirin in vitro (Fig 1J). We found that the organoids of aged mice had less budding and organoid-forming abilities, Aspirin treatment increased these abilities (Fig 1K, 1L, S1F and S1G) and enhanced the differentiation of aged ISC, as shown by real-time qPCR analysis (Fig 1M). Furthermore, the aged mice administered with Aspirin for 3 months also showed alleviation of colon histological symptoms, as decreased crypt width and length (Fig 1N and 1O), which meant that Aspirin also inhibits the ISC differentiation defect and rescue the aged intestinal alterations in vivo. Taken together, these data suggest that Aspirin administration significantly increases aged ISC differentiation and relieves the age-associated intestinal functional decline.

### Aspirin restrains age-associated ISC differentiation defect by enhancing the function of peroxisomes

To identify the mechanism by which Aspirin promotes the differentiation of ISCs during aging, RNA sequencing (RNA-seq) was performed on dissected aged *Drosophila* midguts (40 days under 25 °C) both with and without Aspirin administration (Fig 2A). The transcriptional signature enrichment plot shows that a cluster of peroxisome-related genes was

 

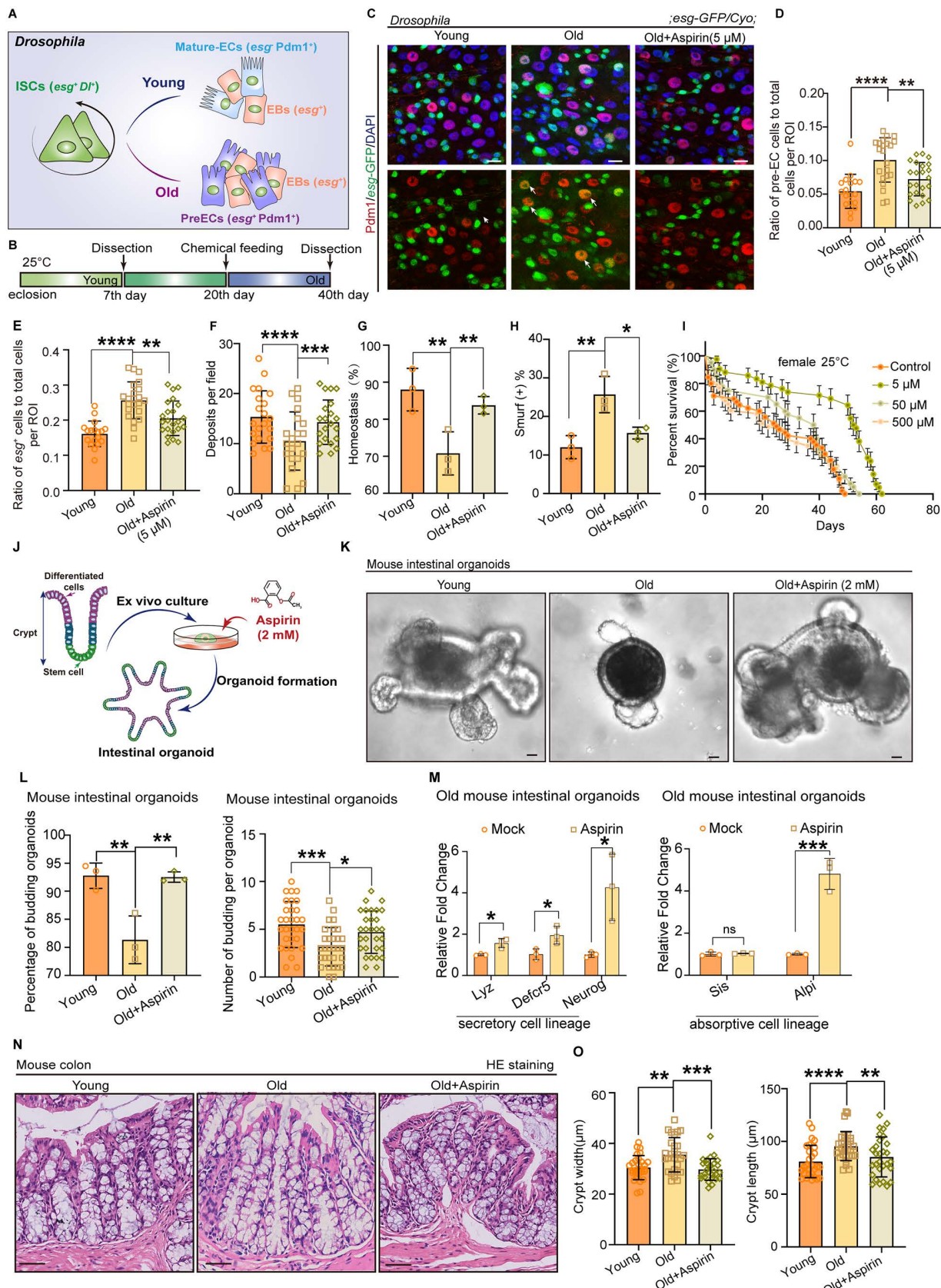

**Fig 1. Aspirin restrains age-associated ISC mis-differentiation and prevents intestine functional decline. (A)** The model of ISC lineages in young and old *Drosophila*. The ISC (Dl+ and *esg*+) in young *Drosophila* produces a new ISC and differentiates into a precursor enteroblast (EB; *esg*+). The post-mitotic EB further differentiates into pre-enterocyte (preEC; *esg*+ and Pdm1+), and then differentiates into a mature enterocyte (ECs; Pdm1+). In aged ISCs, it differentiates into preEC, and weakly continues to differentiate into mature ECs. **(B)** A model illustrating chemical administration with *esg*-GFP *Drosophila* reporter line. The chemicals were fed to *Drosophila* on the 20th day after eclosion. After administration for 20 days, *Drosophila* were dissected for analysis. **(C)** Immunofluorescence images of midguts with or without Aspirin treatment. White arrows indicate preEC. **(D, E)** Quantification of the ration of *esg*+ cells (including ISC and EB cells) and preECs (*esg*-GFP+, Pdm1+cells) in midguts with indicated treatments. **(F)** Quantification of the deposits in indicated treatment. Excretions are quantified in 21 fields for each group of *Drosophila*. **(G)** Quantification of the ration of "Homeostasis" with or without Aspirin supplementation. **(H)** Quantification of the percentage of "Smurf" flies with or without Aspirin supplementation after consuming a non-absorbed food dye. Each sample contains three independent experiments. **(I)** Survival of female W1118 *Drosophila* with and without supplementation of Aspirin after eclosion for 20 days. Each treatment consisted of 50 mated females and the experiment was repeated three times independently. **(J)** The model of mouse intestinal organoid culture. **(K, L)** Representative images, and the quantification of organoid budding and forming ability of young and aged organoids in the presence or absence of Aspirin treatment. **(M)** Real-time qPCR analysis demonstrated that old mouse organoids treated with Aspirin enhanced the ability of aged ISC differentiation. **(N)** Representative images of hematoxylin and eosin (H&E) staining in paraffin-embedded mouse colon sections from young mice, aged mice, and aged mice with Aspirin treatment. **(O)** Average height and width of the crypts in colons from young mice, aged mice, and aged mice with Aspirin treatment. DAPI-stained nuclei (blue). Scale bars represent 10 μm (C), 20 μm (K), 40 μm (N). Error bars represent SDs. Student's *t*-tests, Kruskal–Wallis test, one-way ANOVA, and log-rank test, $*p < 0.05$, $**p < 0.01$, $***p < 0.001$, $****p < 0.0001$, and NS (non-significant) represents $p > 0.05$. Underlying data and statistical analysis in S1 Data.

significantly expressed in aged midguts of *Drosophila* that received Aspirin (Fig 2B and 2C). To determine whether Aspirin regulates peroxisome function to promote ISC differentiation during aging, we first examined the number and function of peroxisomes in ISCs of aged flies. The endogenous peroxisome reporter line Peroxin 10-mCherry (*Pex10-mCherry*) was generated [15] and showed that the abundance of peroxisomes increased dramatically in aged ISCs (ISCs are indicated by *esg-GFP*, and peroxisomes are indicated by *Pex10-mCherry*) (Fig 2D and 2E). However, the peroxisome activity, as indicated by catalase activity assay, was decreased in the ISCs of aged flies (Fig 2F). To test whether the changes to peroxisomes were conserved in mammals, human colon tissues (using the normal para-carcinoma region of young and aged humans) and mouse colons were collected. By performing Western blotting, immunofluorescence staining, and transmission electron microscopy (TEM), we also found a significant abundance with peroxisome (Fig 2G–2K, using PMP70 antibody or "Pex" indicate peroxisomes), and a decrease in peroxisome activity (as indicated by Catalase staining) (Fig 2L) in the crypts of aged mammal colon tissue. In addition, in Aspirin-treated aged mice, we found there was no change in peroxisome abundance, but an increase in peroxisome function in crypts (Fig 2M–2P). These data clearly indicate that the abundance of peroxisomes increases dramatically and that peroxisome function decreases significantly in ISCs during aging, while Aspirin administration may reverse the decline of ISC differentiation by enhancing peroxisomal function.

## PEX5-mediated peroxisomal matrix protein transportation is impaired in aged ISCs

We further explored how Aspirin would enhance the function of peroxisomes. Firstly, the volcano plots and heatmap from the RNA-seq data revealed that Aspirin can up-regulate a cluster of peroxisome function-related genes in aged *Drosophila*, including *acox1*, *acat1*, *acox3*, *abcd1*, and others (Fig 3A and 3B). It is reported that conserved peroxin (*Pex*) genes are responsible for the formation and maintenance of cellular peroxisome populations. In the volcano plots and heatmap, there are only three *Pex* genes significantly increased (including *Pex1, Pex5*, and *Pex6*) after Aspirin administration; these genes mainly function by importing matrix proteins into the peroxisome (Fig 3A–3C). There are two ways for importing peroxisomal matrix proteins in mammal: PEX5 is the principal import receptor, ferrying most proteins into the peroxisome via the C-terminal peroxisomal targeting signal 1 (PTS1) motif, whereas a minor subset bearing N-terminal PTS2 signals is delivered by PEX7—a pathway that has yet to be confirmed in flies [9,29–32] (Fig 3C). Then, we confirmed that there was a significant increase in the expression of *Pex5* in ISCs following Aspirin administration (Fig 3D). Aspirin treatment also increased the expression of luciferase driving by *Pex5-Gal4* (S2A and S2B Fig). Consequently, we speculated that Aspirin may regulate the function of peroxisomes by enhancing PEX5-mediated peroxisomal matrix transport.

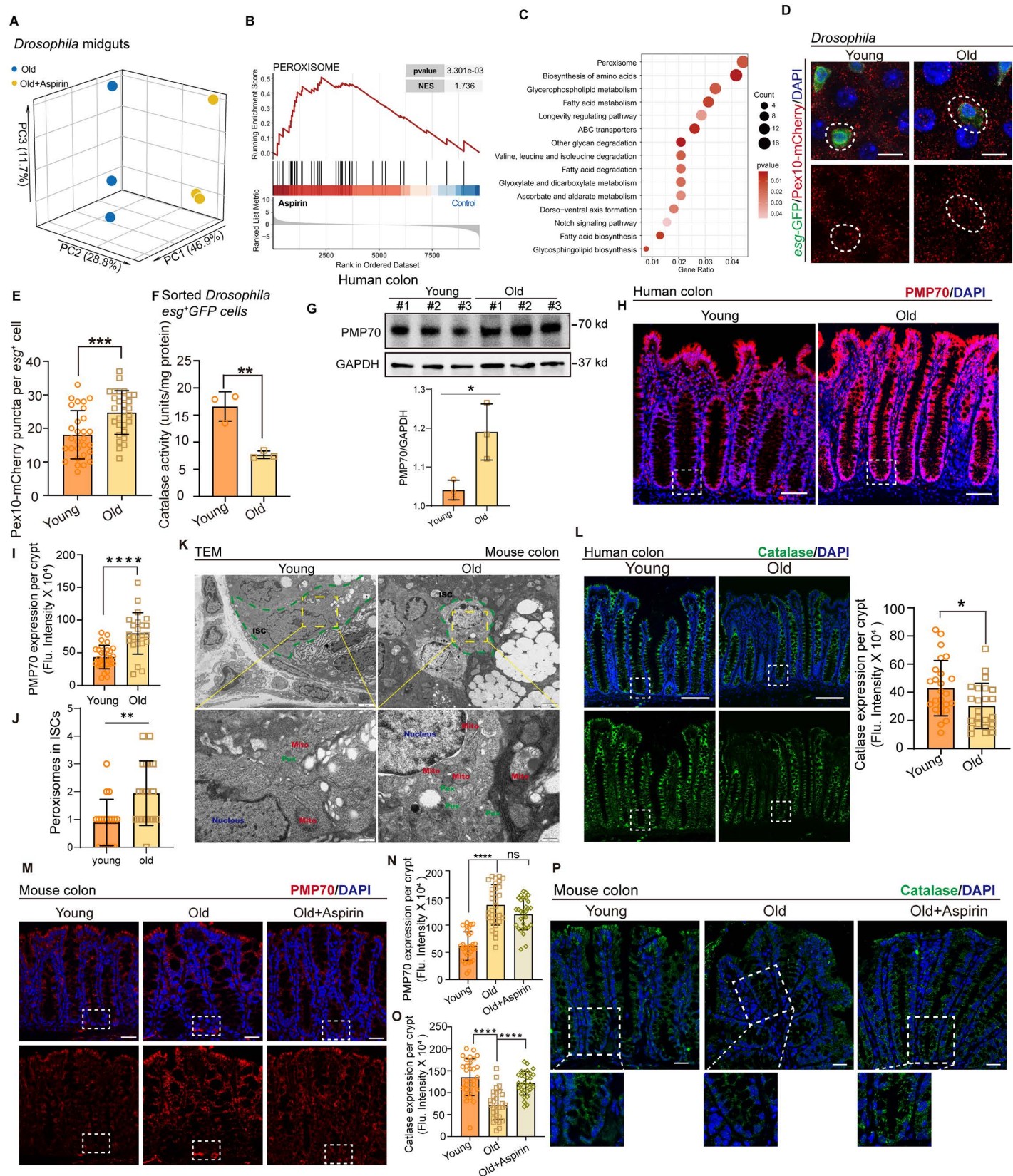

**Fig 2. Aspirin restrains age-associated ISC differentiation defect by enhancing the function of peroxisome. (A–C)** Enrichment plot for transcriptional signature in old *Drosophila* midgut with or without Aspirin treatment. **(D)** Immunofluorescence images showed the abundance of peroxisomes in aged ISCs (as indicated by *esg*-GFP). The peroxisomes were labeled by mCherry (red). **(E)** The quantification of peroxisomes in (D). **(F)** Measurement of Catalase activity with sorted *esg*-GFP cells from young and aged *Drosophila*. n indicates three independent experiments. **(G)** Western blotting result of PMP70 in human normal para-carcinoma colon. Loading controls, α-Tubulin. **(H, I)** Immunofluorescence of PMP70 staining in young and aged human colons. Young and aged Normal para-carcinoma colons were collected. The boxed areas indicate crypts. **(J, K)** TEM from the mouse colon crypt. The green dashed lines: ISC; Pex, peroxisomes; Mito: mitochondrion. **(L)** The Catalase expression in intestinal crypts from young and aged human colons. Young and aged Normal para-carcinoma colons were collected. The boxed areas indicate crypts. **(M, N)** Immunofluorescence of PMP70 staining (red) in mouse intestinal crypts with or without Aspirin treatment. The boxed areas indicate crypts. **(O, P)** Immunofluorescence and statistical analysis of Catalase (green) in mouse intestinal crypts with or without Aspirin treatment. The boxed areas indicate crypts. DAPI-stained nuclei (blue). Scale bars represent 10 μm (D), 40 μm (H, L), 500 nm (K), 25 μm (M, P). Error bars represent SDs. Student's *t* tests, Mann–Whitney test, and one-way ANOVA, *$p$ < 0.05, **$p$ < 0.01, ***$p$ < 0.001, ****$p$ < 0.0001, and NS (non-significant) represents $p$ > 0.05. Underlying data and statistical analysis in S2 Data.

Next, we analyzed the expression of PEX5 in the human colon during aging. And a significant decrease in PEX5 and increase in PEX7 expression were found in aged human colon (Fig 3E–3G) and mouse crypts (Fig 3H and 3I). Meanwhile, we analyzed the expression of acyl-CoA oxidase 1 (ACOX1), which represents peroxisomal matrix proteins imported by PEX5. Western blot analysis indicated that ACOX1 decreased similarly to PEX5 in aged mouse crypts (Fig 3J). To trace PEX5 endogenous expression within the ISCs of aged *Drosophila*, the peroxisome reporter line Peroxin 5-HA (Pex5-HA) was generated using the CRISPR-Cas9 knock-in system (S2C Fig). Immunofluorescence analyses also showed that PEX5 decreased significantly in ISCs (indicated by *esg*-GFP) of aged *Drosophila* (Fig 3K and 3L). Taken together, these results indicated that the decreased expression of PEX5 in aged ISCs was highly conserved.

The peroxisomal GFP-SKL reporter protein (GFP engineered with a C-terminal S-K-L tripeptide sequence, which serves as a peroxisome targeting signal imported by PEX5) was used to visualize the endogenous PTS1 peroxisomal matrix protein. We found that the intensity of GFP-SKL in per peroxisome was decreased dramatically in the aged ISCs of *Drosophila*, which demonstrated that the peroxisomal matrix proteins imported by PEX5 were significantly impeded during aging (Fig 3M). However, Aspirin administration increased GFP-SKL in the peroxisome of aged ISCs, which indicates that Aspirin enhances the function of peroxisomes by enhancing PEX5-mediated peroxisomal matrix transport.

In line with findings in *Drosophila*, aged mice or colonic organoids treated with Aspirin also displayed an increase in PEX5 levels (Fig 3N–3R). To further understand the function of Aspirin on peroxisomal matrix proteins imported by PEX5, mass spectrometry (MS) analysis of peroxisomes was performed in old mouse colon with or without Aspirin treatment (Fig 3S). We showed that the peroxisomal matrix proteins imported by PEX5 significantly increased following Aspirin treatment in aged mice (Fig 3T). Taken together, these findings suggest that PEX5-mediated transport of peroxisomal matrix proteins is impaired in aged ISCs during the aging process; however, Aspirin enhanced this function.

## Aspirin promotes aged ISC differentiation through the activation of PEX5-mediated PTS1 signaling

As demonstrated, the transportation of matrix proteins into peroxisomes was significantly impaired in ISCs during the aging process, while Aspirin increased the levels of PEX5-mediated matrix proteins within the peroxisome and restrained ISC mis-differentiation. To further understand PEX5 function, cell-type-specific RNA interference (RNAi) was used. Firstly, we found that peroxisome-related matrix proteins (indicated by GFP-SKL) were uniformly distributed in the cytoplasm of the normal ISCs, while knocking down *Pex5* in ISCs resulted in GFP-SKL dispersing throughout the entire cell, including the nucleus, without condensed puncta (Fig 4A). Then, we also found a decreased peroxisomal function (indicated by Catalase and ACOX1) in ISCs after *Pex5* knockdown (Fig 4B–4E). Therefore, we conclude that PEX5 downregulation directly leads to peroxisomal impairment. In addition, *Pex5* RNAi in ISCs resulted in a significant accumulation in pre-ECs under 29 °C for 20 days, which mimics the phenotype of aged midguts in *Drosophila* (termed as mimic aging; Fig 4F and 4G). The digestive functions of *Pex5* RNAi flies under the mimic aging condition were further investigated, showed

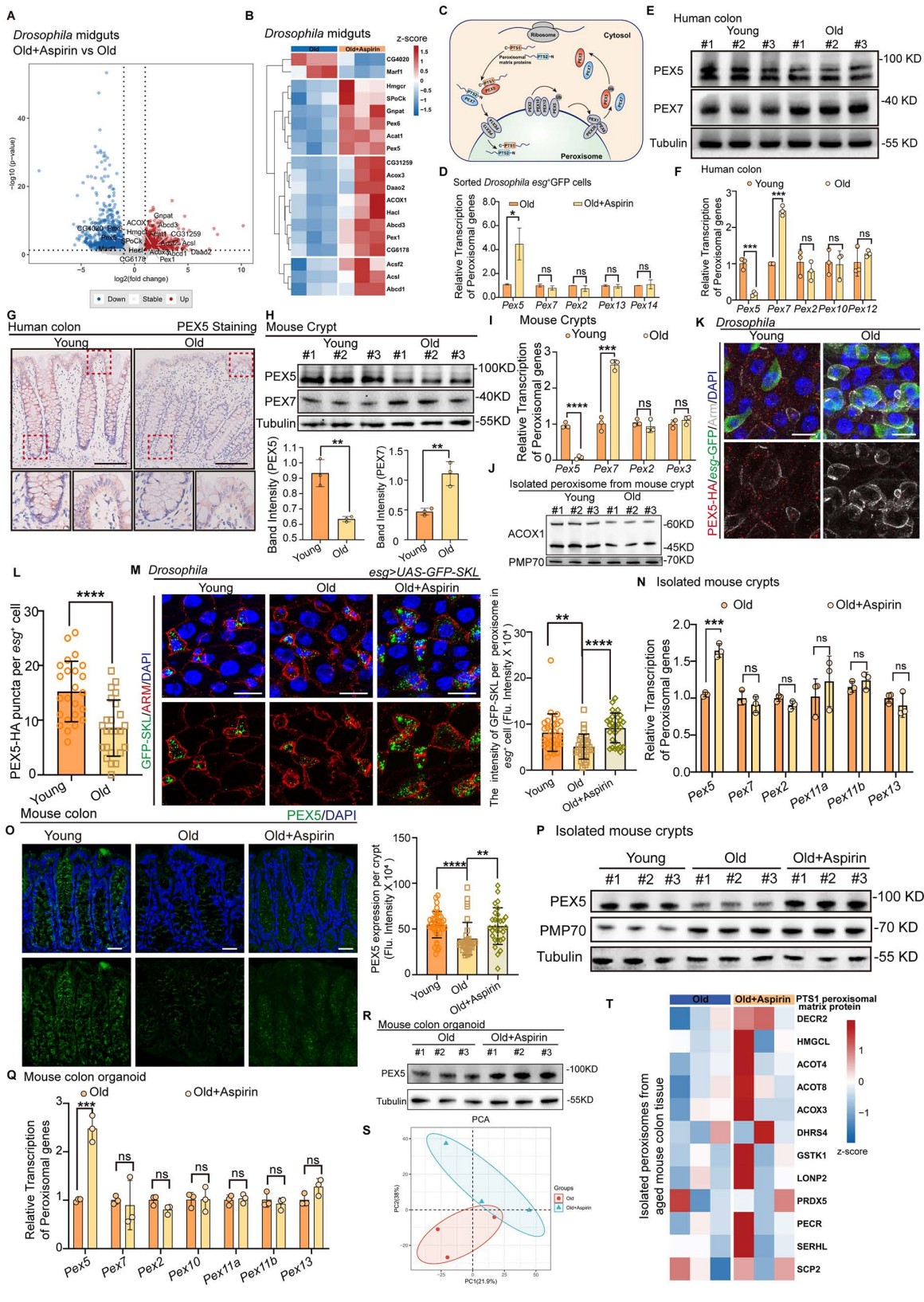

**Fig 3. PEX5-mediated peroxisomal matrix protein transportation is impaired in aged ISCs. (A, B)** The RNA-seq reveals that Aspirin can up-regulate peroxisome-related genes in aged fruit flies, especially increasing the expression of genes associated with PEX5-mediated peroxisomal matrix protein transportation. **(C)** The model of peroxisomal matrix protein transportation. **(D)** Real-time qPCR analysis conducted on sorted ISCs of *Drosophila* further confirmed a significant increase in *Pex5*, following Aspirin administration. **(E–G)** Western Blots, Real-time qPCR, and immunohistochemistry all showed a significant decrease in PEX5 and an increase in PEX7 expression in the aged human colon. **(H, I)** Western Blots, Real-time qPCR showed a significant decrease of PEX5 and an increase of PEX7 expression in aged mouse crypts. **(J)** Western blot analysis indicated that ACOX1 decreased similarly to PEX5 by using isolated peroxisomes from young and old mouse crypts. **(K, L)** Immunofluorescence analyses showed that PEX5 decreased significantly in ISCs of aged *Drosophila. esg*-GFP-labeled ISCs. **(M)** The intensity of GFP-SKL per peroxisome was decreased dramatically in aged ISCs of *Drosophila,* while Aspirin treatment increased its intensity. **(N–P)** Aging mice treated with Aspirin displayed an increase in PEX5 levels in crypts. **(Q, R)** Colonic organoids from aging mice treated with Aspirin also displayed an increase in PEX5 levels. **(S, T)** The Principal component analysis (PCA) of the proteomics. The import of matrix proteins into the peroxisomal lumen via PTS1 was enhanced following Aspirin treatment in aged mice. DAPI-stained nuclei (blue). Scale bars represent 10 μm (G, K, M, O). Error bars represent SDs. Student's *t* tests, Kruskal–Wallis test, and Mann–Whitney test, $*p < 0.05$, $**p < 0.01$, $***p < 0.001$, $****p < 0.0001$, and NS (non-significant) represents $p > 0.05$. Underlying data and statistical analysis in S3 Data.

---

a decline in gastrointestinal acid–base homeostasis (Fig 4H–4J) and excretion (Fig 4K). More importantly, Aspirin did not rescue the ISC differentiation defect (Fig 4F and 4G) and the digestive functions of *Pex5* RNAi flies (Fig 4H–4K). In addition, to further confirm that aspirin enhances ISC differentiation through PEX5 in mammalian systems, we silenced Pex5 in middle-aged mouse colon organoids (S3A–S3C Fig). Then found that *Pex5* knockdown markedly reduced mouse organoid budding, and subsequent aspirin treatment failed to restore its metrics. Therefore, these data indicate that the pro-differentiation effect of aspirin is PEX5-dependent.

## Over-expression of PEX5 in aged flies relieves the intestine functional decline

Since Aspirin restrains the age-associated ISC differentiation defect by promoting the expression of PEX5, it is reasonable to investigate whether the overexpression of PEX5 in aged flies promotes ISC differentiation, relieves the age-associated functional intestine decline. We found that the overexpression of PEX5 from an old age for 10 days significantly decreased age-associated pre-ECs retained in aged midguts (Fig 5A and 5B) and increased the intensity of GFP-SKL in the peroxisome of aged ISCs (Fig 5C and 5D). Furthermore, overexpression PEX5 significantly prevented the deterioration of gastrointestinal acid–base homeostasis (Fig 5E) and excretion (Fig 5F) in aged flies. We also found that the overexpression of PEX5 improved the life span of flies, while the knockdown of *Pex5* shortened its life span when conducted after eclosion of 10 days (Fig 5G and 5H). These results show that overexpression of PEX5 restrains the age-associated ISC differentiation defect and improves the digestive functions of aged flies. Aspirin promotes the differentiation of aged ISCs and enhances digestive functions, specifically through the activation of PEX5-mediated PTS1 signaling.

## Impaired PEX5-mediated PTS1 signaling induces ISC mis-differentiation through modulation VLCFA metabolism

Fatty acids with a chain length exceeding 20 carbon atoms (referred to as VLCFAs) are predominantly oxidized in peroxisomes. Most enzymes involved in the β-oxidation of VLCFAs in peroxisomes are imported by PEX5; this includes ACOX1, which is the primary rate-limiting enzyme [33] (Fig 6A). To investigate whether VLCFAs play a role in the decreased differentiation function of ISCs with aging, targeted quantitative analysis of free fatty acids (FFAs) was conducted by Gas Chromatography–Mass Spectrometry (GC–MS) using mouse colon. Principal component analysis (PCA) revealed a distinct separation between the young, old, and old Aspirin-treated group (Fig 6B). Furthermore, heatmap enrichment, volcano plot analyses, and the bar charts of FFAs demonstrate a significant enrichment of VLCFAs in old mouse colonic tissue, which were shown to significantly decrease after Aspirin treatment (Fig 6C–6E). As we have demonstrated a decrease in ACOX1 expression in peroxisomes isolated from aged mouse colon crypts (Fig 3J), all these findings provide evidence that VLCFAs also accumulated in aged ISCs. To establish that reduced PEX5 in aged ISCs drives VLCFA accumulation, we isolated ISCs with *Pex5* knockdown and quantified FFAs by GC-MS. VLCFAs—namely behenic (C22:0) and lignoceric (C24:0) acids—were markedly elevated, whereas shorter-chain species such as palmitic (C16:0) and stearic (C18:0)

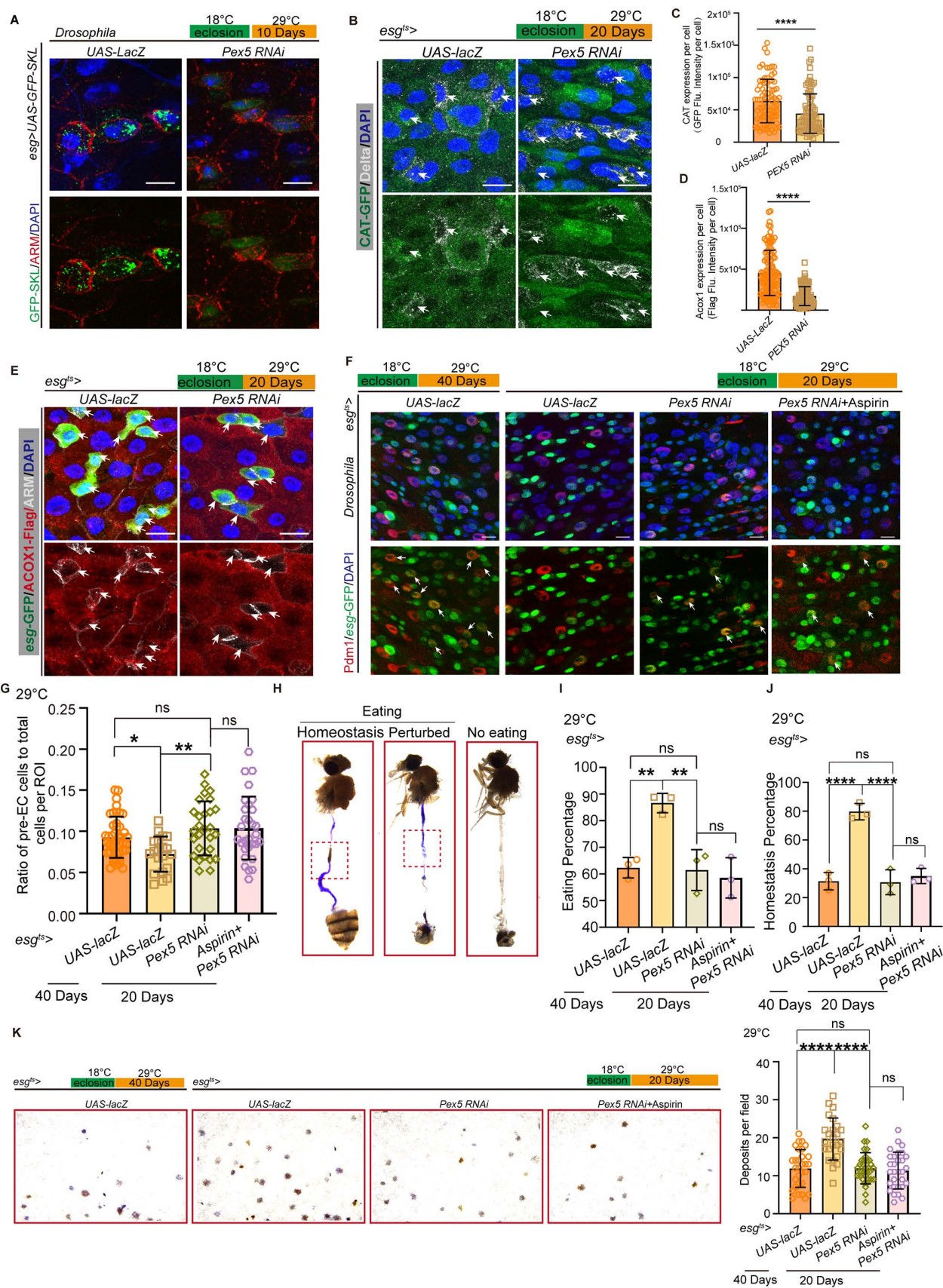

**Fig 4. Aspirin promotes aged ISC differentiation through the activation of PEX5-mediated PTS1 signaling. (A)** Knocking down *Pex5* in ISCs resulted in peroxisome-related proteins (as indicated by GFP-SKL) dispersing throughout the entire cell. **(B–E)** PEX5 downregulation directly leads to peroxisomal impairment, as indicated by decreased expression of ACOX1 and Catalase expression after knocked down *Pex5* in ISCs. **(F)** *Pex5* RNAi in ISCs led to a significant increase of preECs in mimic aging condition, which indicated as 29 °C for 20 days, while the control flies showed no differentiation defect. Aspirin treatment (for 7 days) did not rescue the ISC differentiation defect. **(G)** Quantification of the ratio of esg+ cells (including ISC and EB cells) and preECs (*esg*-GFP+, Pdm1+cells) in midguts as indicated genotypes. **(H–K)** The digestive functions of *Pex5* RNAi flies declined, including excretion, and gastrointestinal acid–base homeostasis. Aspirin treatment (for 7 days) did not rescue the digestive functions of *Pex5* RNAi flies. DAPI-stained nuclei (blue). Scale bars represent 10 μm (A, B, E, F). Error bars represent SDs. Student's *t* tests, Mann–Whitney test, Kruskal–Wallis test, and one-way ANOVA, *$p < 0.05$, **$p < 0.01$, ***$p < 0.001$, ****$p < 0.0001$, and NS (non-significant) represents $p > 0.05$. Underlying data and statistical analysis in S4 Data.

acids remained unchanged (S4A Fig). This selective accumulation confirms a strong link between decreased PEX5 and VLCFA buildup in aged ISCs. Collectively, our findings indicate that aspirin restores ISC differentiation by enhancing PEX5-dependent peroxisomal β-oxidation of VLCFAs.

We further investigated whether the accumulation of VLCFAs regulates ISC differentiation upon aging. When feeding flies with behenic acid (BA, C22-0, typical VLCFAs), it significantly led to an ISC differentiation defect in the young midguts of flies as indicated by the *Pex5* RNAi phenotype (Fig 7A and 7B). Then, feeding old flies with Bezafibrate [34], which reduces VLCFA synthesis through direct inhibition of fatty acid elongation activity, significantly relieved the aged ISC differentiation defect as indicated (Fig 7C and 7D). In addition, we observed that inhibiting the degradation of VLCFAs in the ISCs of flies by *Acox1* RNAi also led to the accumulation of progenitor cell and pre-ECs (Fig 7E and 7F). Reducing VLCFAs generation in aged ISCs by *Elov1* (a VLCFA synthase) [33] RNAi can alleviate aged ISC differentiation defects, as indicated by the *Pex5* overexpression phenotype (Fig 7G and 7H). Therefore, these data demonstrate that the accumulation of VLCFAs in aged ISCs led to an ISC differentiation defect during aging.

Given that our previous research demonstrated that peroxisomes enhance ISC differentiation by enhancing RAB7-dependent late endosome maturation and JAK-STAT-SOX21A signaling [15], it is logical to investigate whether VLCFAs regulate aged ISC differentiation by influencing the expression of SOX21A and late endosome maturation. To determine the effect of VLCFAs on the expression of SOX21A, which is specifically found in ISCs and EBs [35], we conducted VLCFAs and bezafibrate feeding experiments. Firstly, we found that the expression of SOX21A was decreased in aged ISCs, which was consistent with mis-differentiation of aged ISCs. As expected, the treatment of young *Drosophila* with VLCFAs resulted in a decrease in SOX21A expression, while bezafibrate treatment in aged *Drosophila* led to an increase in SOX21A levels (Fig 7I and 7J). Moreover, the forced expression of either SOX21A or RAB7 in ISCs partially corrected the differentiation defects of ISCs in *Acox1* RNAi *Drosophila*, which manifested in a reduction in the accumulation of pre-ECs (Fig 7K and 7L). Therefore, these results indicated that the increased VLCFAs regulated aged ISC differentiation by preventing RAB7-dependent late endosome maturation and JAK-STAT-SOX21A signaling. To further confirm that PEX5 governs aged ISC differentiation by regulating RAB7-dependent late-endosome maturation and JAK-STAT-SOX21A signaling, we specifically knocked down *Pex5* in ISCs. This manipulation markedly reduced the levels of SOX21A, 10XSTAT-GFP, and RAB7-GFP, underscoring PEX5's central role in regulating these (S4B–S4D Fig). Meanwhile, overexpression of either SOX21A or RAB7 ameliorated the differentiation deficiencies caused by *Pex5* RNAi (Fig 7M and 7N). Taken together, these findings indicate that the β-oxidation of VLCFAs, which are downstream of PEX5, governs aged ISC differentiation by modulating RAB7-dependent late endosome maturation and SOX21A expression in the midguts of flies (Fig 7O).

## Discussion

The aging process is complex and multifaceted, characterized by a gradual decline in function, and recognized as a major risk factor for nearly all age-related diseases [7]. Beyond genetic factors, scientific evidence suggests that nutritional and pharmacological interventions can help to counteract aging [24–26]. The intestinal epithelium serves as a dynamic barrier,

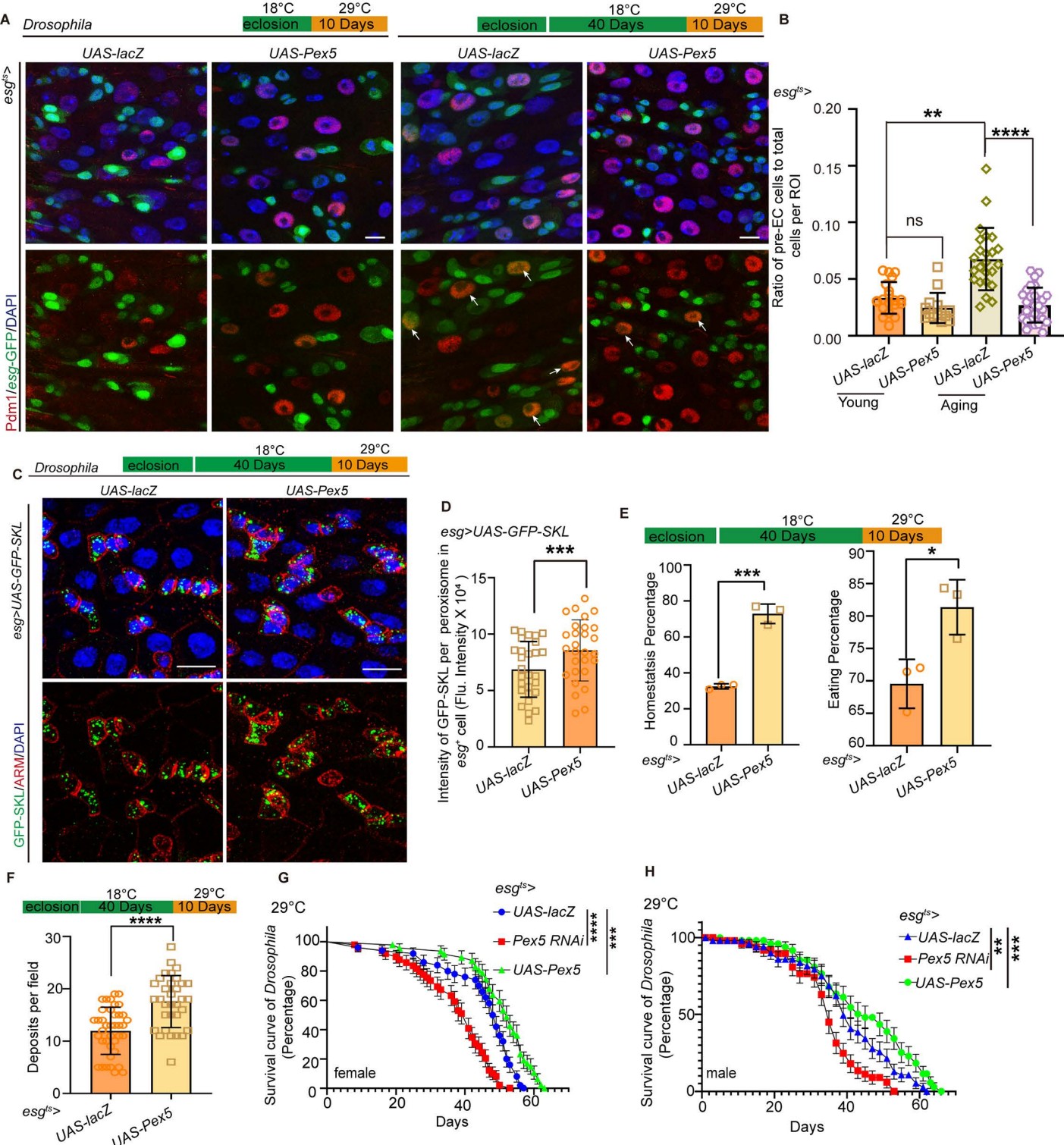

**Fig 5. Over-expression of PEX5 in aged flies relieves the intestine functional decline. (A, B)** Overexpression of PEX5 started at intermediate age (40 days, 18 °C) significantly decreased age-associated preECs detain in aged midguts. **(C, D)** Overexpression of PEX5 started at intermediate age (40 days, 18 °C) significantly increased the intensity of GFP-SKL per peroxisome in aged ISCs. **(E, F)** Overexpression of PEX5 starting at an intermediate

age significantly prevented further deterioration of gastrointestinal acid–base homeostasis and excretion in aged flies. **(G, H)** Overexpression of PEX5 improved the lifespan of flies, whereas knockdown of PEX5 resulted in a shortened lifespan. Each treatment consisted of 50 mated females, and the experiment was repeated three times independently. DAPI-stained nuclei (blue). Scale bars represent 10 μm (A, C). Error bars represent SDs. Student's *t* tests, Mann–Whitney test, Kruskal–Wallis test, and log-rank test, $*p < 0.05$, $**p < 0.01$, $***p < 0.001$, $****p < 0.0001$, and NS (non-significant) represents $p > 0.05$. Underlying data and statistical analysis in S5 Data.

integrating signals from metabolites, commensal microbiota, immune responses, and stressors throughout aging [36]. These changes impact the physiological responses of the entire organism, affecting stem cell behavior, plasticity, and environmental responses, ultimately influencing intestinal homeostasis. Recent research has focused on the communication between organs to understand how gut health relates to overall organismal health. Our study represents a significant breakthrough, showing that Aspirin can mitigate the aging of ISCs and effectively address diseases related to intestinal aging, with the age-associated dysregulation of peroxisome function playing a crucial role in stem cell aging.

Aspirin is a non-steroidal anti-inflammatory drug that was first introduced to the field of medicine in 1898. It is a derivative of acetylated salicylic acid and is also known as acetylsalicylic acid. Aspirin plays a pivotal role in alleviating fever and pain, exerting anti-thrombotic effects, and treating inflammatory and rheumatic conditions. It has been demonstrated to have a growing number of functions in diverse areas such as cardiovascular and cerebrovascular health, diabetes management, and in combination with anti-cancer medications [37–39]. In the context of intestinal health, numerous studies have highlighted the potential of Aspirin to reduce the incidence of CRC and positively impact the aging process in the intestine [40,41]. However, the precise mechanisms by which Aspirin operates in these contexts, particularly when considering its effects on ISCs, have not yet been fully elucidated. In this study, we demonstrated that Aspirin delays intestinal aging by improving the differentiation deficiencies observed in aged ISCs, both in *Drosophila* and mouse. Although there is a significant discrepancy in the aspirin concentrations, which may ascribe species differences. In addition, it enhanced the expression of PEX5 and led to a reduction in VLCFAs within aged ISCs. This study marks the first reported instance of the anti-aging effects of Aspirin through the regulation of organelle function, suggesting that peroxisomes could potentially serve as a promising target for Aspirin treatment in various other diseases.

Recent studies suggest that peroxisomes are critical mediators of cellular responses to various forms of stress, including oxidative stress [11,42], hypoxia [43], starvation [44], cold exposure [45], and noise [13]. However, the way by which peroxisomes change during aging, especially in ISCs, is deeply unknown. In this study, we unveiled an increased abundance of peroxisomes in aged ISCs, accompanied by a decline in their functionality. From our findings, the decrease in peroxisomal function is related to low PEX5 expression in aged ISCs, which results in a decrease in the import of metabolic enzymes into the peroxisome, accompanied by a reduction in VLCFA β-oxidation. Ultimately, this leads to the mis-differentiation of aged ISCs. Meanwhile, increasing the expression of PEX5 in aged ISCs enhances the differentiation capacity of ISCs and the intestinal tract barrier. Therefore, these results prove that the dysfunction of peroxisomes caused by PEX5 is a leading cause of aged ISC mis-differentiation. Yet, whether the increased compensatory peroxisomes lead to an increase in differentiation disorders of aging ISCs necessitates further study.

VLCFAs are relatively rare species, comprising only 1%–2% of total fatty acids [46], and they remain poorly understood relative to other lipids. VLCFAs have unique properties [19], including the ability to stabilize curved membranes, to generate ATP via β-oxidation, and to modify various proteins (e.g., GPI anchors) post-translationally. However, the accumulation of VLCFAs can cause necroptosis by disrupting the membrane [22], causing membrane saturation [47] and the unfolded protein response, in addition to inducing tumor growth [21]. An inability to use VLCFAs may lead to an increased occurrence of X-linked ALD [48,49] and CRC [23]. In the current study, we first discovered that VLCFAs are elevated in the aged intestine, therefore inhibiting RAB7-dependent late endosome maturation and SOX21A expression, which led to the mis-differentiation of aged ISCs. Therefore, this study establishes a novel relationship between VLCFAs and aged ISCs, providing a foundation for exploring the mechanisms underlying differentiation disorders in aging ISCs.

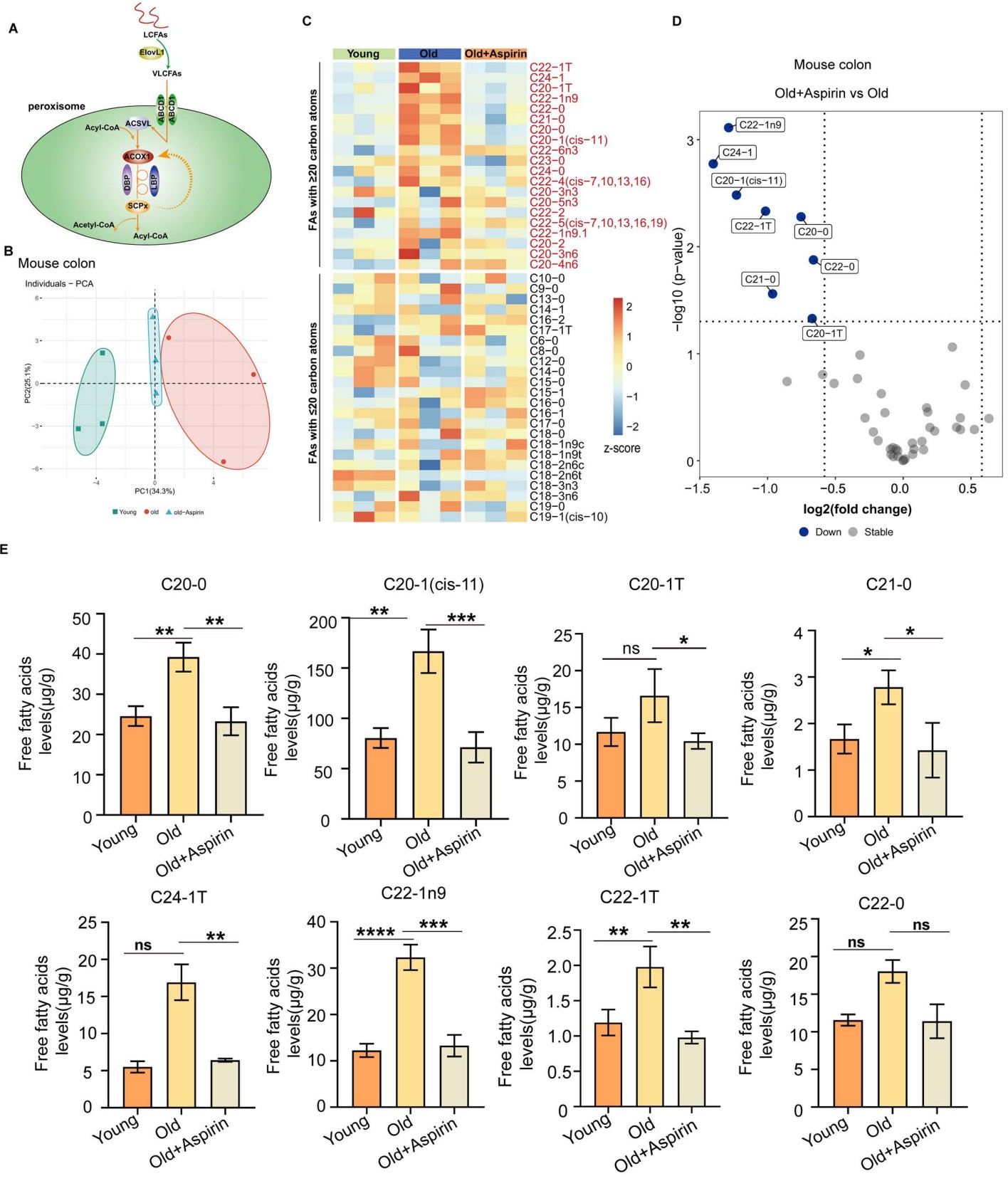

**Fig 6. A substantial accumulation of VLCFAs was found in aged ISCs. (A)** Model of peroxisomal β-oxidation of VLCFAs. **(B)** The Principal component analysis (PCA) of free fatty acids. **(C)** Heatmap enrichment of quantified FFAs in indicated groups. **(D, E)** Bar charts and volcano plots show representative FFA levels in indicated groups. Error bars represent SDs. one-way ANOVA, $*p < 0.05$, $**p < 0.01$, $***p < 0.001$, $****p < 0.0001$, and NS (non-significant) represents $p > 0.05$. Underlying data and statistical analysis in S6 Data. .

## Materials and methods

### *Drosophila* stocks and husbandry

All *Drosophila* stocks used in this study are listed below with complete fly genotypes provided in S1 Table. The following lines were obtained from the Bloomington *Drosophila* Stock Center (BDSC): $w^{1118}$ (BDSC# 3605), *UAS-lacZ* (BDSC# 8529), *UAS-lacZ* (BDSC# 3956), *UAS-GFP-SKL* (BDSC# 28881), *UAS-Sox21a* (BDSC# 68156), *UAS-Rab7-CA* (BDSC# 50785), *UAS-Rab7-GFP* (BDSC# 42706), *10XStat-GFP(BDSC# 26200)*, *CAT-GFP(BDSC# 51546)*, *UAS-Luciferase* (BDSC# 35788), *Elovl RNAi* (BDSC# 50710), *ACOX1 RNAi* (BDSC# 68109). *PEX5 RNAi* (TH02247.N) was obtained from the TsingHua Fly Center (THFC). The *esg-GFP/CyO* and *esg-Gal4* were kindly provided by Allan Spradling. The *esg^{ts}-Gal4* line: *esg-Gal4, UAS-GFP, tub-Gal80^{ts}/CyO* was a gift from Benjamin Ohlstein. The transgenic *Drosophila* lines *SOX21A-HA, UAS-PEX5-HA, Pex10-mCherry, Pex5-HA, ACOX1-Flag, and Pex5-Gal4* were constructed in our laboratory. Fly lines were backcrossed into the $w^{1118}$ background (6 generations) and sibling populations were derived from crosses of $w^{1118}$.

*Drosophila* was raised on a standard cornmeal-agar medium under conditions of 25 °C and 65% humidity, following a 12:12 light and dark cycle, unless otherwise specified. For temperature-sensitive control of Gal4-driven gene expression, flies were first reared at 18 °C to minimize driver activity and then maintained at 29 °C to induce gene activation. The experiments only included mated female flies.

### Mouse models

The animal procedures in this study were conducted in accordance with the guidelines provided by the Committee on the Ethics of Animal Experimentation at West China Hospital. Ethical approval was granted (Permit Number: 202203170). Male C57BL/6J mice were used in this study and were purchased from GemPharmatech in Nanjing, China, and Aniphe Biolaboratory Mice of varying ages were utilized in the study, with 3-month-old mice classified as young and 18–21-month-old mice categorized as aged. The specific age of the mice used in each experiment will be delineated in the respective figure legends or methods. The mice were housed in the SPF facility at the Animal Experimental Center of West China Hospital, Sichuan University. They were kept under a 12-hour light-dark cycle, with a relative humidity of 40–50% and a temperature of $22 \pm 2$ °C. The mice had ad libitum access to food and water, unless otherwise stated. Prior to the experiment, the mice were adaptively fed for 7 days.

### Human samples

Human colon tissues from patients were obtained from the Division of Gastrointestinal Surgery, West China Hospital, Sichuan University. These tissues were harvested from 30 CRC patients and are classified as adjacent normal tissue (refer to Table 1 for a comprehensive list of patient details). These patients were contacted to obtain verbal informed consent. This study protocol adhered to the ethical principles of the 1975 Helsinki Declaration and received approval from the Human Ethics Committee of West China Hospital, Sichuan University, in 2023 (Approval numbers:20231214).

### Generation of knock-in and transgenic fly strains

To examine the expression of endogenous PEX5 in *Drosophila*, we created a *PEX5-HA* knock-in line. First, we inserted the sgRNA sequence into the PMD18T vector that contains the U6 promoter. Next, we amplified the U6

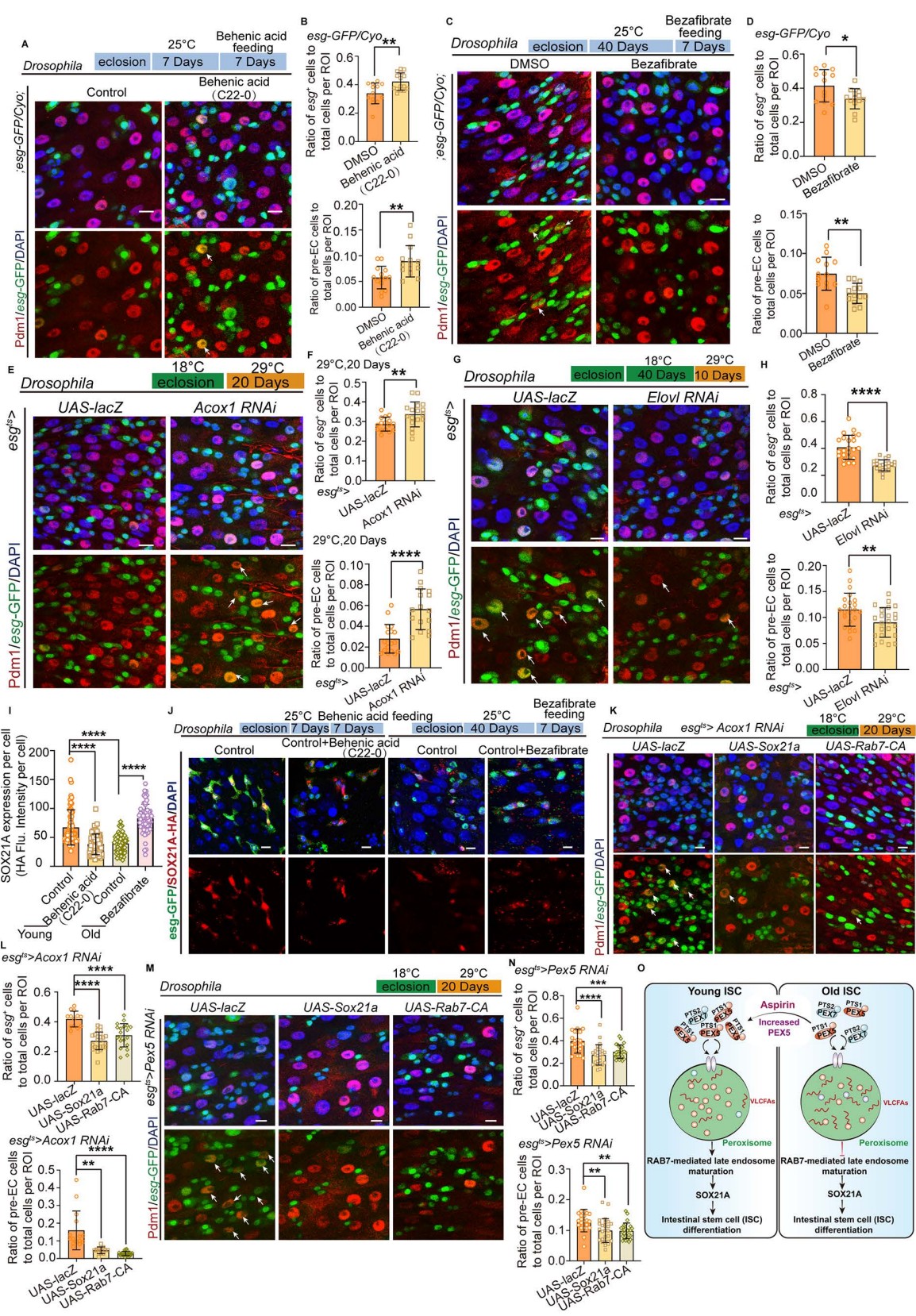

**Fig 7. Impaired PEX5-mediated PTS1 signaling induces ISC mis-differentiation through modulation VLCFAs metabolism. (A, B)** Feeding flies with VLCFAs (Behenic acid, C22-0) significantly led to ISC differentiation defect. **(C, D)** Feeding old flies with Bezafibrate significantly relieved the ISC differentiation defect. **(E, F)** Inhibiting the degradation of VLCFAs in midguts of flies by *ACOX1* RNAi led to the accumulation of progenitor cells and preECs under mimic aging conditions. **(G, H)** Knocking down of *Elovl* in ISC could inhibit ISC differentiation defects in aged midguts. **(I, J)** Treatment of young *Drosophila* with VLCFAs resulted in a decrease in SOX21A expression, while Bezafibrate treatment in old *Drosophila* led to an increase in SOX21A levels. **(K, L)** Enforced expression of either SOX21A or RAB7-CA in ISCs partially corrected the differentiation defects in ISCs in *ACOX1* RNAi *Drosophila*, manifested by a reduction in the accumulation of preECs. **(M, N)** The overexpression of either SOX21A or RAB7-CA also ameliorated the differentiation deficiencies in ISC in *Pex5* RNAi midguts. **(O)** The model of the current study. DAPI-stained nuclei (blue). Scale bars represent 10 μm (A, C, E, G, J, K, M). Error bars represent SDs. Student's *t* tests, one-way ANOVA, Mann–Whitney test, and Kruskal–Wallis test, $*p < 0.05$, $**p < 0.01$, $***p < 0.001$, $****p < 0.0001$, and NS (non-significant) represents $p > 0.05$. Underlying data and statistical analysis in S7 Data.

**Table 1. The demographic of all participants.**

| Participant characteristics | Young | Old |
|---|---|---|
| Number of patients | 15 | 15 |
| Age, years (median, range) | 26 (21–30 ) | 70 (62–79) |
| Gender (male/female) | 7/8 | 9/6 |

promoter and sgRNA from the PMD18T vector through PCR. Then, we transferred them into the PCR8 vector using the Golden Gate assembly method. These products were subsequently recombined into the attB vector through LR recombination. We integrated the HA tag into the PEX5 gene through homologous recombination in the PASK vector. Lastly, the *Drosophila* embryos were microinjected, a procedure performed by Qidong Fungene Biotechnology, in Beijing, China.

We also created the *PEX5-RA-Gal4* transgenic lines. Briefly, the pUASTattB-gal4 vector (from Qidong Fungene Biotechnology) which has the attB site and miniwhite marker was used as the backbone. 2,137 bp of DNA immediately upstream of the transcription start site of gene pex5-RA was cloned into *Not I/Nde I* sites of pUASTattB-gal4. The injection was performed by Qidong Fungene Biotechnology and screening integration flies.

To overexpress PEX5 in *Drosophila*, we generated *UAS-PEX5-HA* transgenic lines. First, we cloned cDNAs of Pex5 into the pEntry-3×HA vector and then into the pTW vector. After confirming the plasmids through sequencing, UniHuaii Corporation (Zhuhai, China) performed the injections.

Primers used for these constructions are listed in S2 Table.

## Immunofluorescence microscopy for *Drosophila* midguts

*Drosophila* midguts were dissected in cold PBS and fixed for 30 min using a 4% EM-grade paraformaldehyde solution mixed with n-Heptane. The tissues were then washed twice with methanol for 5 min each. Next, they were permeabilized twice in 0.1% PBST (PBS containing 0.1% Triton X-100), for 10 min each time. The tissues were incubated overnight at 4 °C with primary antibodies. Following this, they were washed three times with 0.1% PBST before incubating with secondary antibodies for 2 hours at room temperature. After the secondary antibody incubation, the guts were washed three times with 0.1% PBST for 10 min each time. Place the tissue on a slide and cover it with Antifade Mounting Medium with DAPI. The dilutions and sources of the antibodies used are listed as follows: anti-GFP (1:1,000; Abcam), anti-HA (1:1,000; Cell Signaling Technology), anti-Armadillo (1:50; DSHB), anti-Delta antibody (1:75; DSHB), anti-mCherry (1:1,000; Invitrogen), anti-Pdm1 (1:200), and Alexa Fluor secondary antibodies (1:2,000). The Leica TCS-SP8 confocal microscope was used to acquire all immunofluorescence images. For each set of experiments, images were acquired as confocal stacks using the same settings.

 

## Behenic acid (BA), Bezafibrate, and Aspirin administration in *Drosophila*

Behenic acid (BA, Sigma), C22-0, typical VLCFAs, was dissolved and mixed into regular food at a concentration of 2.5% (w/v). The Bezafibrate (MCE) was dissolved in DMSO and then added to the regular food to achieve a concentration of 0.4 μM. Newly eclosed flies were collected and placed into vials containing the regular food for 7 days or 40 days, and then fed with BA or Bezafibrate for 7 days. Aspirin (Aladdin) was dissolved in water and then added to the regular food to achieve a concentration of 5 μM. Newly eclosed flies were collected and placed into vials containing the regular food for 20 days, and then they were transferred to food supplemented with the forementioned drugs for 20 days. The food was changed every two days until the midguts were dissected.

## Fluorescence-activated cell sorting

*Drosophila* guts were dissected in cold DEPC-PBS. After the dissection, the midguts were incubated in a Trypsin-EDTA solution at 25 °C for 40 min with gentle shaking. The dissociated samples were then centrifuged at 3,000 rpm for 20 min, filtered through 40 μm cell strainers, and sorted using flow cytometry (FACS Aria III sorter, BD Biosciences, USA). Specifically, *esg*-GFP+ cells from the midgut of flies treated with or without Aspirin were sorted. The fluorescence threshold was set using the *w1118* midgut as a reference for sorting. Approximately 180,000 *esg*-GFP+ cells were collected for each of the three biological replicates.

## Bromophenol blue assay

In accordance with a previously established protocol [50], the Bromophenol blue assay was used to assess the acid–base balance in the gut. 2% Bromophenol blue sodium (pH indicator, Sigma, B5525) was added to the standard food in a vial, and the food's surface was punctured multiple times with a pipette tip to ensure complete absorption. After a 24-hour feeding period, images of the dissected guts were promptly captured.

## Fly excretion measurement

After 2 hours of starvation, an assay on fly excretion was conducted. The flies were placed into vials containing food with 2% bromophenol blue sodium, while the interiors of the vials were wrapped with chromatography paper. After observing the flies for 24 hours, the deposits of excretion on the paper were photographed and measured. Each group in the study consisted of 15 flies.

## 'Smurf' assay

FD&C Blue #1 dye was added to the standard cornmeal-agar food at a concentration of 2.5% (w/v). After 2 hours of starvation, the flies were fed with the FD&C Blue #1 dyed food for 12 hours and then observed. Flies that exhibited a visible blue color outside their digestive tract, known as Smurf (+), were counted.

## Lifespan experiments

For the survival tests under normal conditions, we collected 50 mated female or male flies of each treatment (*esgts > UAS-lacZ*, *esgts > Pex5 RNAi*, and *esgts > UAS-Pex5*). These flies were housed at 18 °C for 10 days, and then equally distributed into five vials containing standard food and housed at 29 °C. To test the protective effect of Aspirin, we collected 50 mated female or male flies of each group fed with regular food for 20 days, and then equally distributed them into five vials containing 5 μM Aspirin food, housed at 25 °C. The number of living flies was counted and transported to new vials every 2 days. The experiment was repeated three times independently.

## Luciferase assays

The transcriptional regulation effect of drug treatment on target genes can be assessed by measuring the activity of the luciferase enzyme. To confirm the effect of Aspirin on Pex5, we conducted a luciferase assay using the reporter

fly: *Pex5-Gal4 > UAS-Luciferase*. We compared the luciferase activity in the guts of flies that were fed with Aspirin and those without. The Firefly Luciferase Reporter Gene Assay Kit (Beyotime) was used to measure luciferase activity. Fifteen female midguts were dissected in cold PBS and homogenized in 50 µL lysis buffer. The sample extracts were then centrifuged at 13,000g for 10 min at 4 °C. The supernatant was transferred to a 96-well plate for detection. Equal amounts of firefly luciferase detection reagent were added to each well, and the relative light unitswere measured after incubation.

## Quantification of GFP-SKL fluorescence intensity for per peroxisome

The fluorescence intensity was quantified using the fluorescence analysis tool in Leica confocal software. For each ISC cell, the total fluorescence intensity of GFP-SKL was measured, and the number of GFP-SKL puncta within the cell was counted. Then, the average fluorescence intensity per GFP-SKL punctum was calculated by dividing the total fluorescence intensity by the number of GFP-SKL puncta.

## RNA-seq and data analysis

About 30 female flies for each biological replicate were dissected in cold PBS and frozen immediately on liquid nitrogen. The isothiocyanate-alcoholphenyl-chloroform was used to collect total RNA for midguts. Then, total RNA was sent to Berry Genomics Corporation (Beijing, China) for sequencing on the NovaSeq 6000 platform (Illumina, San Diego, CA, USA). Quality control was performed using FastQC (version 0.11.8). The raw RNA-seq data with read lengths of 150 bp were aligned to the *Drosophila* reference genome (Ensembl BDGP6 release-110). The aligned reads (sam files) were transferred to bam files and sorted using feature Counts (version 1.5.1). Gene expression levels in different samples were determined using DESeq2 (version 1.40.2). Differential expression was verified with adjusted <0.05 and log2 (fold change) >1 or log2 (fold change) <−1. The cluster Profiler software package version 4.8.3 was used to perform functional annotation and enrichment analysis of differentially expressed genes. The Complex Heatmap software package version 2.13.1 was used to visualize expression patterns among samples using heatmap representation. The PCA was performed by the statistics function prcomp within R (v4.3.1) (www.r-project.org).

## Isolation of mouse colonic crypts and organoid culture

Mouse colonic crypts were isolated as described previously with some modifications [51]. Mouse colons were flushed with cold PBS and cut longitudinally. Villi were scraped off gently with a glass slide. The remaining tissue was cut into 1–2 mm sections. Colonic pieces were washed several times until the supernatant was clear and incubated in 5 mM EDTA-PBS for 90 min in the ice box on a rocking platform. The lysis buffer was removed carefully and replaced with cold PBS containing 0.1% bovine serum albumin (BSA, Gibco). The collected crypts were filtered with a 70-µm mesh to remove villous fragments. Finally, filtered crypts were used for real-time qPCR, western blotting analyses, or organoid culture.

For organoid culture, collected crypts were resuspended in Advanced DMEM/F12 medium (Gibco, 12634010), and the quality of the suspensions was assessed using a microscope. Unless otherwise specified, crypts were grown in a mixture of Advanced DMEM/F12 and GlutaMAX (31980030, 12491015) supplemented with N2 (1:100, Thermo 17502048), B27 (1:50, thermo17504044), Wnt3a (1:10, MLB J2-001), Noggin 100 ng/ml (1:1000, Peprotech, 250–38), R-Spondin1,500 ng/ml (1:500, Peprotech, 120–38), N-acetyl-ʟ-cysteine 1 mM (Sigma, A7250), Recombinant Mouse EGF Protein 50ng/ml (1:100,000, Peprotech, 315-09), SB202190 (1:1,000, Sigma, S7067), Y-27632 10 µM (APExBIO, A3008), A8301 0.5 µM (Tocris, 2939) and Matrigel (BD bioscience, 356230) and incubate at 37 °C in a humidified incubator at 5% $CO_2$. Organoids were exposed to 2 mM Aspirin (Aladdin), and the crypt medium was changed every three days. Images of the

organoids were captured under a 20× microscope. Reagents were typically solubilized in dimethyl sulfoxide (DMSO) for experimental use unless otherwise specified. Noggin and R-Spondin1 dried protein powder were dissolved in a 5% alginate solution.

## siRNA-mediated gene silencing in mouse organoids

Two days before electroporation, mouse organoids were passaged and cultured in organoid medium. To promote the formation of cystic hyperproliferative crypts, the medium was supplemented with 10 μM Y27632 (HY-10071, Med Chem Express), 10 μM CHIR99021 (HY-10182, Med Chem Express), and 10mM nicotinamide (N108087, Aladdin) [52]. The siRNA sequences were as follows: *Pex5* siRNA sense strand, 5′-GGAUCCUAAGCACAUGGAA(dT) (dT)-3′, and anti-sense strand, 5′-UUCCAUGUGCUUAGGAUCC (dT) (dT)-3′ (antisense). As previously reported methods [53], organoids were dissociated into single cells using 0.25% Trypsin-EDTA (Gibco 25200072) at 37 °C. Then, isolated single cells were resuspended in BTXpress electroporation buffer (BTX) along with the siRNA and subjected to electroporation using the Gene Pulser Xcell Total System (Bio-Rad, 1652660) under parameters optimized as described. Following electroporation, the single cells were cultured in the ENR medium supplemented with 500ng/mL R-spondin-1 (final concentration, 1 μg/mL), 10μM Jagged-1 peptide (Anaspec, AS-61298), and 10% Afamin-Wnt-3A serum-free conditioned medium (MBL) to facilitate the generation of organoids.

## Real-time qPCR

Total RNA from human colon samples, isolated mouse colonic crypts, mouse colonic organoids, or sorted *Drosophila esg*-GFP+ cells was extracted using RNA-easy Isolation Reagent (Vazyme, R701-01) according to the manual. After measuring the RNA concentration and quality, the RNA was reverse transcribed into cDNA using *Evo M-MLV* RT Mix Kit (AG, AG11728). Real-time qPCR was completed using ChamQ Universal SYBR qPCR Master Mix (Vazyme, Q712-02). Relative expression of target genes was normalized to *Rp49* in *Drosophila* or *Gapdh* in mice and human, and estimated with the $2^{-\Delta\Delta CT}$ method. Primers used for real-time qPCR are listed in S2 Table.

## Aspirin administration in mice

In the aged group treated with Aspirin, 17-month-old male C57BL/6J mice were fed with drinking water with 1mg/ml (5.5mM) Aspirin for 3 months. Conversely, in the control aged group, the same aged male C57BL/6J mice were provided with drinking water without Aspirin for the 3-month period.

## Immunofluorescence and immunohistochemistry for human and mammalian tissues

The colon tissues were fixed in Neutral Buffered, followed by dehydration, paraffin embedding, and sectioning. Bake the sections in the oven at 65 °C for 30min and apply biotransparent for 30min to remove the paraffin. The sections were washed sequentially with a gradient of ethanol, starting with anhydrous ethanol, then 95% ethanol, then 90% ethanol, and finally 80% ethanol, each for 5min. Following this, a 5-min rinse with double-distilled water (ddH2O) was performed. The sections were incubated with 3% $H_2O_2$ at room temperature for 30min, followed by washing with ddH2O. Citrate was used for microwave antigen repair, followed by washing with PBS for 5min twice. Then blocking with 5% serum for 30min, primary antibody (anti-PMP70 1:500, anti-catalase 1:500, anti-PEX5 1:400) was added and incubated at 4 °C overnight. After washing with PBS for 10min, three times, secondary antibodies were added and incubated at room temperature for 90min. Samples were then washed and imaged. For immunohistochemistry, after washing with PBS for 10min, three times, biotin-conjugated secondary donkey anti-rabbit or anti-mouse antibodies were added and incubated at 37 °C temperature for 1h. The Vectastain Elite ABC immunoperoxidase detection kit (Vector Labs, PK-6101) followed by DAB Horseradish Peroxidase Color Development Kit (Beyotime, P0203) was used for visualization.

## Histological analysis

The colon tissues were fixed in a 10% formalin solution for 24 h, and then underwent a process of dehydration and paraffin embedding to prepare sections. Hematoxylin and eosin (H&E) staining was used to visualize the crypt structures in the colon slices. Statistical analysis was performed to measure the length and width of the colonic crypts.

## Peroxisome Isolation

In order to better investigate the role of peroxisomes in the aging process and the effects of the drug Aspirin on peroxisomes, we utilized the Peroxisome Isolation Kit and a density gradient centrifugation method with iodixanol [OptiPrep] to isolate peroxisomes from mouse colonic crypts.

(1) Sample preparation: Harvest the colon tissue from the old and old + Aspirin groups, rinse with cold PBS, and add 4 volumes of 1× Peroxisome Extraction Buffer per gram of colon tissue, followed by homogenization using a cryogenic tissue grinder.

(2) Preparation of Crude Peroxisomal Fraction (CPF): Centrifuge the homogenate at 1,000$g$ for 10 min at 4 °C to remove nuclei and other cellular debris. Further centrifuge at 2,000$g$ for 10 min at 4 °C to remove heavy mitochondria. Then, centrifuge at 25,000$g$ for 20 min at 4 °C. Aspirate off the supernatant and resuspend the pellet in a minimal volume of 1× Peroxisome Extraction Buffer to obtain the Crude Peroxisomal Fraction (CPF).

(3) Isolation of Peroxisomes on a Density Gradient: Following the protocol, dilute the CPF to form an OptiPrep concentration of 22.5%. Then, sequentially layer the prepared OptiPrep concentrations of 27.5%, 22.5% (diluted CPF), and 20% OptiPrep solution. Centrifuge for 1.5 hours at 100,000$g$. Withdraw the bottom layer containing the purified peroxisomes, followed by rapid freezing in liquid nitrogen for subsequent proteomic analysis and other experiments.

## Shotgun proteomics and data analyses

To further explore the changes in peroxisomes, we subjected the isolated peroxisomes from the intestines of old mice and old mice treated with Aspirin to Shotgun Proteomics analysis. The isolated peroxisomes underwent protein extraction, enzymatic digestion into peptides, and liquid chromatography–tandem mass spectrometry (LC–MS/MS) analysis. Briefly, the samples were lysed using an SDT lysis buffer SDT (4%SDS, 100 mM Tris-HCl, 1 mM DTT, pH7.6) for protein extraction, followed by protein quantification using the BCA kit. Protein digestion was performed by trypsin. The digest peptides of each sample were desalted on C18 Cartridges concentrated by vacuum centrifugation and reconstituted in 0.1% formic acid. Each sample was then separated using an HPLC liquid chromatography system. The mobile phase A was a 0.1% formic acid aqueous solution, and phase B was a 0.1% formic acid acetonitrile solution (with acetonitrile at 84%). The chromatographic column was equilibrated with 95% phase A before the sample was loaded for separation through the analytical column. Post-chromatographic separation, the samples were analyzed using Q Exactive mass spectrometer coupled with an Easy nLC system. The raw MS data were processed for database searching and quantitative analysis using the software MaxQuant. The Shotgun Proteomics analysis was completed by the Institute of New Life Sciences. The protein quantification method is label-free quantification (LFQ). The LFQ values of the samples are normalized by Z-score, and the protein content of the samples is displayed in the form of a heat map.

## Detection of free fatty acid in mouse colonic tissue

The targeted FFA assay was performed using gas chromatography–mass spectrometry (GC–MS). Tissue samples from the colons of mice categorized as young, old, and old treated with Aspirin were individually weighed, each approximately 100 mg, for the analysis. Fatty acid extraction from the colonic tissue was performed as previously described [16]. Sample derivatives were analyzed using a GC-EI-MS system (GC, Agilent 8890; MS, 5977B System) with SIM

scanning mode. The analytical conditions were the same performed as previously described [16]. The above fatty acid detection tests were performed by MetWare using the Agilent 8890-5977B GC-MS platform. Unsupervised PCA was performed by the statistics function prcomp within R (v4.3.1) (www.r-project.org). The data was unit variance scaled before unsupervised PCA. Differential metabolites selected between groups were determined by absolute $\log_2$ (fold change) and showed by volcano plot.

## Catalase activity analysis

We assessed catalase enzyme activity in human colon tissue, mouse colonic crypts, and sorted *Drosophila esg⁺ GFP* using the Catalase Assay Kit (Beyotime, Cat# S0051). The above tissues were lysed with RIPA Lysis Buffer (Beyotime, P0013C) to prepare the samples for analysis. Subsequently, the protein content was quantified using the BCA Protein Assay Kit (Beyotime, P0011). We formulated a hydrogen peroxide solution and the chromogenic working solution, added the latter to the samples, and initiated a timed color development reaction. The residual hydrogen peroxide in the samples was measured by absorbance at A520 with a microplate reader, allowing us to calculate the catalase activity based on the protocols provided by the manufacturer.

## Transmission electron microscopy

The human colon was fixed with a 3% glutaraldehyde solution, followed by post-fixed with 1% osmium tetroxide. It was then infiltrated with acetone and resin mixtures, before being embedded in epoxy resin (Epon812) at 60 °C for 36 hours. For microscopic examination, ultrathin sections were prepared and stained with uranyl acetate and lead citrate. These sections were subsequently observed under a JEM1400 TEM. Intestinal stem cells (ISCs) were identified by their small nuclei and their proximity to the basal membrane of the gut epithelium. Meanwhile, peroxisomes were identified through their crystalline morphology and the presence of a granular matrix that stained positively for DAB.

## Western blotting analyses

To monitor changes in peroxisome-related proteins at the protein level, we performed western blot experiments on human colon, isolated mouse colonic crypts, mouse colon-isolated peroxisome, and mouse colonic organoids. The above samples were frozen with liquid nitrogen and homogenized with tissue lysate in RIPA lysis buffer containing 1% cocktail and 1% PMSF, then placed on ice for 60 min for thorough lysis. The tissue lysates were centrifuged at 12,000 rpm for 15 min at 4 °C, and the supernatants were collected. Protein mixed with loading buffer was boiled for 5 min and analyzed using 10%–12% SDS-PAGE. Then, the protein was transferred onto polyvinylidene difluoride (PVDF) membranes. The PVDF membranes were blocked with 5% non-fat milk in TBST for 60 min and then incubated overnight at 4 °C with primary antibodies (anti-PMP70, 1:500; anti-PEX5, 1:1,000; anti-PEX7, 1:1000; anti-ACOX1, 1:1000; anti-ACAA1, 1:1,000; anti-α-Tubulin, 1:1,000). After being washed three times with TBST at room temperature (10 min each), the PVDF membranes were incubated with the secondary antibody at room temperature for 90 min. Detection was performed using an ECL chemiluminescence detection kit (Vazyme, E422-01). Images were captured using the ChemiDoc XRS+ system and analyzed with Image Lab (v5.1, Bio-Rad). The band gray value statistics of Western Blotting were calculated using ImageJ software, following the previous methods [16].

## Quantification and statistical analysis

Data analysis for the detection of FFAs, as well as for shotgun proteomics and RNA-seq, is thoroughly detailed in the methods section of the relevant section. Excepting that, all statistical analysis was performed with GraphPad Prism 8. Data are presented as the means ± SD of at least three independent tests unless otherwise noted. The life span assays were tested for significance with a log-rank test. Statistical significance was assessed with the unpaired two-tailed

Student's *t* test unless otherwise specified. In all cases, a $p < 0.05$ was used to test for statistical significance. N.S., not significant. All experiments were independently repeated for at least three times.

## Supporting information

**S1 Fig. Aspirin restrains age-associated ISC differentiation defect by enhancing the function of peroxisomes.**
**(A)** Treatment of *Drosophila* with 50 or 500 μM aspirin failed to restrain the age-associated ISC differentiation defect.
**(B)** Excretion of *Drosophila* treated with Bromophenol blue. **(C)** Representative images and quantification of *Drosophila* midguts treated with the pH indicator Bromophenol blue. There are three conditions: Homeostasis, a well-defined acidic (yellow colored) in copper cell region (CCR), and anterior midgut (AM) and posterior midgut (PM) is basic (blue colored); "Perturbed" the acidic region is lost and the whole gut is basic; "No eating", no Bromophenol blue showed in the midguts. **(D)** Representative images of "Smurf" flies after consuming a non-absorbed food dye. **(E)** Aspirin treatment extended the life span of male *Drosophila*. **(F, G)** Varying Aspirin concentrations (2 μM/20 μM/200 μM/2 mM/20 mM) on mouse intestinal organoids culturing. Error bars represent SDs. Student's *t* tests, one-way ANOVA, Kruskal–Wallis test, and log-rank test, *$p < 0.05$, **$p < 0.01$, ***$p < 0.001$, ****$p < 0.0001$, and NS (non-significant) represents $p > 0.05$. Underlying data and statistical analysis in S8 Data.
(DOCX)

**S2 Fig. PEX5-mediated peroxisomal matrix protein transportation is impaired in aged ISCs. (A)** The strategy used for constructing the endogenous *Drosophila* Pex5-Gal4 knock-in line. **(B)** Aspirin treatment also increased the expression of luciferase driving by Pex5-Gal4. **(C)** The strategy used for constructing the endogenous *Drosophila* Pex5-HA knock-in line. Error bars represent SDs. Student's *t* tests, *$p < 0.05$, **$p < 0.01$, ***$p < 0.001$, ****$p < 0.0001$, and NS (non-significant) represent $p > 0.05$. Underlying data and statistical analysis in S9 Data.
(DOCX)

**S3 Fig. Aspirin promotes aged ISC differentiation through the activation of PEX5-mediated PTS1 signaling. (A)** RT-qPCR analysis of Pex5 in mouse intestine organoids after transfection with siPex5 ($n = 3$ biologically independent mice per group). **(B)** Representative images showing the growth of organoids over 7 days. The organoids were transfected with control silencing RNA (siControl) and Pex5 silencing RNA (siPex5). **(C)** Quantification of organoids buds in each group. Error bars represent SDs. Scale bars represent 75 μm (A). Student's *t* tests and one-way ANOVA, *$p < 0.05$, **$p < 0.01$, ***$p < 0.001$, ****$p < 0.0001$, and NS (non-significant) represents $p > 0.05$. Underlying data and statistical analysis in S10 Data.
(DOCX)

**S4 Fig. Impaired PEX5-mediated PTS1 signaling induces ISC mis-differentiation through modulation VLCFA metabolism. (A)** VLCFA detection in ISCs after Pex5 RNAi. **(B–D)** A significant reduce of SOX21A, 10× STAT-GFP, and RAB7-GFP expression in ISCs after Pex5 RNAi. Scale bars represent 10 μm (B, C, D). Error bars represent SDs. Student's *t*-tests and Mann–Whitney test, *$p < 0.05$, **$p < 0.01$, ***$p < 0.001$, ****$p < 0.0001$, and NS (non-significant) represents $p > 0.05$. Underlying data and statistical analysis in S11 Data.
(DOCX)

**S1 Text. Key resources table.**
(DOCX)

**S1 Table. Full *Drosophila* genotypes as they appear in each figure panel.**
(XLSX)

**S2 Table. Primers for real-time qPCR and constructs.**
(XLSX)

**S1 File. Raw data of free fatty acids by GC–MS.**
(RAR)

**S1 Data. Underlying data for Fig 1.**
(XLSX)

**S2 Data. Underlying data for Fig 2.**
(XLSX)

**S3 Data. Underlying data for Fig 3.**
(XLSX)

**S4 Data. Underlying data for Fig 4.**
(XLSX)

**S5 Data. Underlying data for Fig 5.**
(XLSX)

**S6 Data. Underlying data for Fig 6.**
(XLSX)

**S7 Data. Underlying data for Fig 7.**
(XLSX)

**S8 Data. Underlying data for S1 Fig.**
(XLSX)

**S9 Data. Underlying data for S2 Fig.**
(XLSX)

**S10 Data. Underlying data for S3 Fig.**
(XLSX)

**S11 Data. Underlying data for S4 Fig.**
(XLSX)

**S1 Raw Images. Raw Images for Fig 2G.**
(TIF)

**S2 Raw Images. Raw Images for Figs 3E, 3H, 3J, 3P, and 3R.**
(TIF)

## Acknowledgments

We thank BDSC and Tsinghua Fly Center for fly strains and DSHB for antibodies.

## Author contributions

**Conceptualization:** Xiaoxin Guo, Gang Du, Juanyu Zhou, Haiyang Chen.

**Data curation:** Xiaoxin Guo.

**Formal analysis:** Xiaoxin Guo.

**Funding acquisition:** Haiyang Chen.

**Investigation:** Xiaoxin Guo, Juanyu Zhou.

**Methodology:** Xiaoxin Guo, Gang Du, Juanyu Zhou, Fang Fu, Yu Yuan, Xingzhu Liu, Haiou Chen.

**Resources:** Haiyang Chen.

**Supervision:** Haiyang Chen.

**Validation:** Xiaoxin Guo.

**Visualization:** Xiaoxin Guo, Gang Du, Juanyu Zhou, Bo Gong.

**Writing – original draft:** Xiaoxin Guo, Gang Du, Haiyang Chen.

**Writing – review & editing:** Xiaoxin Guo, Juanyu Zhou, Qianyi Wan, Haiyang Chen.

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
