## [Editor Report · Decision Letter 0]

14 Jan 2025

Dear Dr Chen,

Thank you for submitting your manuscript entitled "Aging-Related Peroxisomal Dysregulation Disrupts Intestinal Stem Cell Differentiation via Modulating Lipid Homeostasis" for consideration as a Research Article by PLOS Biology.

Your manuscript has now been evaluated by the PLOS Biology editorial staff as well as by an academic editor with relevant expertise and I am writing to let you know that we would like to send your submission out for external peer review.

Once your full submission is complete, your paper will undergo a series of checks in preparation for peer review. After your manuscript has passed the checks it will be sent out for review. To provide the metadata for your submission, please Login to Editorial Manager (https://www.editorialmanager.com/pbiology) within two working days, i.e. by Jan 16 2025 11:59PM.

Kind regards,

Ines

--

Ines Alvarez-Garcia, PhD

Senior Editor

PLOS Biology

---

## [Decision Letter · Decision Letter 1]

27 Mar 2025

Dear Dr Chen,

Thank you for your patience while your manuscript entitled "Aging-Related Peroxisomal Dysregulation Disrupts Intestinal Stem Cell Differentiation via Modulating Lipid Homeostasis" was peer-reviewed at PLOS Biology. Your manuscript has been evaluated by the PLOS Biology editors, an Academic Editor with relevant expertise, and by one independent reviewer. In this case, the Academic Editor has provided as well reviewer level comments.

The reviews are attached below. As you will see, the reviewer and Academic Editor find the work potentially interesting, but they have also raised a substantial number of important concerns that would need to be addressed before we can consider the manuscript for publication. Reviewer 1 suggests additional experiments that should be performed to confirm the conclusions, asks for clarifications on the conservation of the mechanisms across species, for direct evidence to confirm that PEX5 overexpression reduces VLCFA levels, a dose-response curve to back up some of the observations, and quantification of some of the experiments among other issues. The Academic Editor has similar concerns and raises concerns regarding the connection between PEX5 and VLCFA metabolism that would require additional experiments. In addition, s/he asks you to explore whether PEX5 downregulation directly leads to the observed peroxisomal impairment, to confirm the link between peroxisomal function and the signalling pathways during ISC differentiation, and to show if upregulation of SOX21A or RAB7 can rescue the differentiation defects observed in aged ISC.

Based on the specific comments, it is clear that a substantial amount of work would be required to meet the criteria for publication in PLOS Biology. However, we would be open to inviting a comprehensive revision of the study that thoroughly addresses all the comments. Given the extent of revision that would be needed, we cannot make a decision about publication until we have seen the revised manuscript and your response to the reviewers' comments. Your revised manuscript would need to be seen by the reviewers again, but please note that we would not engage them unless their main concerns have been addressed.

We appreciate that these requests represent a great deal of extra work, and we are willing to relax our standard revision time to allow you 6 months to revise your study. Please email us (plosbiology@plos.org) if you have any questions or concerns, or envision needing a (short) extension.

**IMPORTANT - SUBMITTING YOUR REVISION**

3. Resubmission Checklist

a) *PLOS Data Policy*

b) *Published Peer Review*

d) *Blurb*

Please also provide a blurb which (if accepted) will be included in our weekly and monthly Electronic Table of Contents, sent out to readers of PLOS Biology, and may be used to promote your article in social media. The blurb should be about 30-40 words long and is subject to editorial changes. It should, without exaggeration, entice people to read your manuscript. It should not be redundant with the title and should not contain acronyms or abbreviations. For examples, view our author guidelines: https://journals.plos.org/plosbiology/s/revising-your-manuscript#loc-blurb

Sincerely,

Ines

--

Ines Alvarez-Garcia, PhD

Senior Editor

PLOS Biology

Reviewers' comments

Rev. 1:

In their manuscript, Guo et al. reveal how aging impairs intestinal stem cell (ISC) function through peroxisomal dysfunction and dysregulated lipid metabolism. Using Drosophila and mouse intestinal organoids, the study demonstrates that reduced PEX5 expression in aged ISCs leads to impaired import of peroxisomal matrix proteins and the accumulation of very long-chain fatty acids (VLCFAs). The authors identify RAB7-dependent endosome maturation and SOX21A as downstream regulators controlling ISC differentiation. Notably, aspirin restores ISC lineage fidelity by enhancing PEX5-mediated peroxisomal β-oxidation of VLCFAs, suggesting a potential therapeutic strategy to counteract age-related intestinal decline. Although the novelty of this work is partially limited by previous studies—such as the reported role of aspirin in regulating intestinal stem cell (ISC) differentiation via Wnt signaling inhibition (PMID: 32971322), the established link between peroxisomal import dysfunction and aging (PMID: 32523050), and the documented function of very-long-chain fatty acids (VLCFAs) as signaling molecules in peroxisome homeostasis and intestinal repair (PMID: 32243783)—this study consolidates a critical pathway governing aging-associated ISC differentiation. Nevertheless, several key concerns must be addressed before the manuscript can be considered for publication, as detailed below:

Since this study used both drosophila and mammalian systems, the authors should clarify whether the claimed pathway is conserved or not, e.g. whether drosophila has PTS2? Is there any difference between the preEC accumulation in old drosophila and budding defect in mouse intestine organoids?

The study primarily focuses on aspirin's role in improving ISC differentiation via the PEX5 pathway in Drosophila models but lacks validation in mammalian systems (e.g., mouse or human cells). This limits the generalizability and clinical relevance of the findings.

While the data suggest a correlation between decreased PEX5 levels and VLCFA accumulation, direct evidence (e.g., PEX5 overexpression directly reducing VLCFA levels) is lacking.

There is a significant discrepancy in the aspirin concentrations used between Drosophila (5 µM) and mouse intestinal organoids (2 mM). The rationale for this difference should be clarified.

Why does only 5 µM aspirin enhance fly survival (Fig. 1i), while higher concentrations (50 µM, 500 µM) show no effect? A dose-response curve or mechanistic explanation is needed to address this observation.

In Fig. 2g, the increase in PMP70 appears marginal. Grayscale quantification (e.g., band intensity analysis) should be provided to confirm the significance of the results.

In Fig. 3m, the method for quantifying GFP-SKL fluorescence intensity (e.g., normalization per peroxisome) is not described, making it difficult to interpret the data.

In Fig. 1i and 5g, the number of experimental replicates and sample size per group are not stated, which compromises the assessment of statistical power.

Academic Editor’s comments

This manuscript investigates the mechanisms of intestinal stem cell (ISC) aging using Drosophila and mouse colon organoid models. The authors present evidence suggesting that age-related decline in PEX5 expression compromises peroxisomal matrix protein import in ISCs, which is associated with peroxisomal dysfunction and very long-chain fatty acid (VLCFA) accumulation. Their findings indicate that elevated VLCFA levels may disrupt ISC differentiation by interfering with RAB7-dependent late endosomal maturation and JAK-STAT-SOX21A signaling. The authors also report that aspirin treatment appears to restore proper ISC lineage commitment by promoting PEX5-mediated peroxisomal β-oxidation of VLCFAs. While these results point to peroxisomal function and lipid metabolism as potential regulators of ISC aging, the current manuscript would benefit from additional experimental evidence to strengthen the proposed causal relationships and fully support the conclusions.

Substantial revision is recommended to address these limitations.

Major Comments:

1. The authors demonstrate that aspirin treatment can restore the diminished PEX5 levels observed in aged mice and colon organoids. They also show that aged mouse colon tissues exhibit elevated VLCFA concentrations, which are reduced following aspirin administration. However, the mechanistic connection between PEX5 and VLCFA metabolism requires more robust investigation through genetic approaches. Additional experiments involving PEX5 manipulation would strengthen the proposed relationship between these two factors.

2. While the study demonstrates reduced PEX5 expression in aged intestinal stem cells concurrent with peroxisomal dysfunction, the causal relationship between these phenomena requires further experimental validation. Additional mechanistic studies would be valuable to establish whether PEX5 downregulation directly leads to the observed peroxisomal impairment.

3. The manuscript proposes that peroxisomal β-oxidation of VLCFAs, regulated by PEX5, influences ISC differentiation through modulation of RAB7-dependent late endosome maturation and the JAK-STAT-SOX21A pathway. However, this mechanistic framework lacks sufficient supporting evidence or relevant citations from the literature. Additional experimental data or references are needed to establish the proposed connection between peroxisomal function and these signaling pathways in ISC differentiation.

4. Further investigation is warranted to determine whether upregulation of SOX21A or RAB7 could rescue the differentiation defects observed in aged intestinal stem cells. This experimental approach would help establish the functional significance of these factors in age-related ISC differentiation impairment.

5. The quality of the Western blot is poor, the data is overexposed, and the changes mentioned by the authors cannot be discerned, e.g., Figure 2G, 3E, 3J. This insufficient data quality undermines the conclusions drawn from these figures.

6. When describing the results of Western blot and certain immunofluorescence images (e.g., Figure 2O), the manuscript uses qualitative terms like "increased" or "decreased" without providing statistical analysis. It is crucial to include statistical results to support these qualitative descriptions.

7. The manuscript lacks data explaining how Aspirin was identified as a potential candidate. It is crucial to provide evidence or data from the screening process that led to the selection of Aspirin for further investigation. Please include detailed information on the screening process, such as the criteria used, the number of compounds tested, and the specific data that highlighted Aspirin as a promising candidate.

8. In Figure 3S, mass spectrometry analysis of peroxisomes was performed using colon tissues from aged mice, both with and without Aspirin treatment. However, the data displayed considerable heterogeneity between individual samples, and there are no annotations in the figure legend to clarify these observations. To enhance the reliability and interpretability of these data, I recommend performing unsupervised principal component analysis (PCA), which could help account for this variability and further strengthen the conclusions.

9. In Figure 1J, the concentration of Aspirin used to treat mouse intestinal organoids (2 mM) is significantly higher than the 5 µM used in Drosophila. This discrepancy needs to be addressed and explained. Additionally, it would be valuable to investigate whether there is a dose-response effect of Aspirin on mouse intestinal organoids.

10. The study only used female Drosophila. It is important to consider whether gender differences might influence the experimental outcomes. Further investigation into the potential gender-specific effects would strengthen the study's applicability.

11. The manuscript reports that overexpression of Pex5 extends the lifespan of Drosophila. Given the relatively short lifespan of fruit flies, it would be interesting to explore whether similar genetic interventions could have comparable effects in mice. Additionally, considering Aspirin's widespread use, it would be valuable to investigate whether its administration can extend the lifespan of mice.

12. While fatty acid oxidation predominantly occurs in mitochondria, very long-chain fatty acids (VLCFAs) are also known to be metabolized in peroxisomes. In Figure 6, fatty acid detection analyses of mouse intestinal tissues suggest the accumulation of VLCFAs in aging cells. However, the data presented do not address the subcellular localization of these fatty acids. Therefore, the statement "the peroxisomal β-oxidation of VLCFAs" lacks sufficient supporting evidence, as the intracellular compartmentalization of the fatty acids remains unclear.

13. This study primarily investigates the role of fatty acid metabolism in intestinal aging in mice. However, no comprehensive analysis of other lipid classes (e.g., glycerides, phospholipids) is presented. Given the narrow focus on fatty acid oxidation, the term "Lipid Homeostasis" in the title of the manuscript may not be fully appropriate, as it implies a broader scope of lipid regulation that is not addressed in the current study.

Minor Comments:

1. The labeling in Figure S1D is unclear. It is recommended to clearly indicate whether Aspirin was administered or not above the images to avoid confusion.

2. In Figure 2J, it is not clear whether the structures shown are indeed colonic crypts. Additionally, the lack of statistical results weakens the interpretation of the data. Providing clear identification and statistical analysis would enhance the credibility of the findings.

3. The bar chart in Figure 6E lacks significance statistics comparing the two groups. It is essential to include these statistics to validate the differences observed.

4. In Figure 7A, the indication of positive cells is inaccurate. Correcting this would improve the clarity and accuracy of the figure.

5. In Figure 5A, the quality of the representative figure is poor, it does not correspond to the statistics, and the arrows are inaccurately indicated.

---

## [Decision Letter · Decision Letter 2]

27 Oct 2025

Dear Dr Chen,

Thank you for your patience while we considered your revised manuscript entitled "Aging-Related Peroxisomal Dysregulation Disrupts Intestinal Stem Cell Differentiation via Modulating Very Long-Chain Fatty Acid Metabolism" for publication as a Research Article at PLOS Biology. This revised version of your manuscript has been evaluated by the PLOS Biology editors, the Academic Editor and the original reviewer.

Based on the reviews (below), we are likely to accept this manuscript for publication, provided you satisfactorily address the remaining points raised by the Academid Editor. Please also make sure to address the data and other policy-related requests stated below my signature.

In addition, we would like you to consider a suggestion to improve the title:

"Aging-Related Peroxisomal Dysregulation Disrupts Intestinal Stem Cell Differentiation through alterations of Very Long-Chain Fatty Acid oxidation"

We expect to receive your revised manuscript within two weeks.

*Published Peer Review History*

*Press*

Sincerely,

Ines

--

Ines Alvarez-Garcia, PhD

Senior Editor

PLOS Biology

Fig. 1D-I, L, M, O; Fig. 2A-C, E-G, I, J, L, N, O; Fig. 3A, B, D, F, H, I, L-O, Q-T; Fig. 4C, D, G, I, J, K; Fig. 5B, D-H; Fig. 6B-E; Fig. 7B, D, F, H, I, L, N; Fig. S1A, E, O; Fig. S2B; Fig. S3A, C and Fig. S4A-D

CODE POLICY

DATA NOT SHOWN?

Please note that per journal policy, we do not allow the mention of "data not shown", "personal communication", "manuscript in preparation", "unpublished" or other references to data that is not publicly available or contained within this manuscript. Please either remove mention of these data or provide figures presenting the results and the data underlying the figure(s).

Reviewers' comments

Rev. 1:

The authors have adequately addressed all of my concerns. The revisions have led to significant improvements in the manuscript.

Academic Editor's comments

The author has largely answered my concerns, but there are a few other comments, which I believe will make the manuscript more convincing.

1.Inter-group comparisons involve >2 conditions, yet no omnibus test (e.g., one-way ANOVA or Kruskal-Wallis) was applied; instead, only pairwise t-tests are reported. This approach inflates the type-I error rate and is therefore inappropriate. Please replace the repeated t-tests with an ANOVA (or its non-parametric equivalent) followed by a valid post-hoc multiple-comparison procedure.

2.The assumption of normality must be explicitly verified or justified before any parametric test, including t-tests, is used. Otherwise, non-parametric alternatives should be adopted.

3.The heatmaps lack a title for the color legend, such as 3B, 3T, and 6C, which makes it unclear what the color scale represents (e.g., expression level, fold change, or z-score). It is recommended to add an appropriate legend title to clarify the meaning of the color gradient.

---

## [Editor Report · Decision Letter 3]

25 Nov 2025

Dear Dr Chen,

Thank you for the submission of your revised Research Article entitled "Aging-Related Peroxisomal Dysregulation Disrupts Intestinal Stem Cell Differentiation through alterations of Very Long-Chain Fatty Acid oxidation" for publication in PLOS Biology. On behalf of my colleagues and the Academic Editor, Jing Qu, I am delighted to let you know that we can in principle accept your manuscript for publication, provided you address any remaining formatting and reporting issues. These will be detailed in an email you should receive within 2-3 business days from our colleagues in the journal operations team; no action is required from you until then. Please note that we will not be able to formally accept your manuscript and schedule it for publication until you have completed any requested changes.

PRESS

Sincerely, 

Ines

--

Ines Alvarez-Garcia, PhD

Senior Editor

PLOS Biology
